# Multi-domain translation between single-cell imaging and sequencing data using autoencoders

Karren Dai Yang[1,5], Anastasiya Belyaeva[1,5], Saradha Venkatachalapathy[2,3], Karthik Damodaran[2], Abigail Katcoff[1], Adityanarayanan Radhakrishnan[1], G. V. Shivashankar[2,3,4] & Caroline Uhler [1✉]

The development of single-cell methods for capturing different data modalities including imaging and sequencing has revolutionized our ability to identify heterogeneous cell states. Different data modalities provide different perspectives on a population of cells, and their integration is critical for studying cellular heterogeneity and its function. While various methods have been proposed to integrate different sequencing data modalities, coupling imaging and sequencing has been an open challenge. We here present an approach for integrating vastly different modalities by learning a probabilistic coupling between the different data modalities using autoencoders to map to a shared latent space. We validate this approach by integrating single-cell RNA-seq and chromatin images to identify distinct sub-populations of human naive CD4+ T-cells that are poised for activation. Collectively, our approach provides a framework to integrate and translate between data modalities that cannot yet be measured within the same cell for diverse applications in biomedical discovery.

---

[1] Massachusetts Institute of Technology, Cambridge, MA, USA. [2] Mechanobiology Institute, National University of Singapore, Singapore, Singapore. [3] ETH Zurich and Paul Scherrer Institute, Villigen, Switzerland. [4] FIRC Institute of Molecular Oncology (IFOM), Milano, Italy. [5]These authors contributed equally: Karren Dai Yang, Anastasiya Belyaeva. ✉email: cuhler@mit.edu

Recent evidence has highlighted the importance of the 3D organization of the genome to regulate cell-type-specific gene expression programs[1,2]. High-throughput and high-content single-cell technologies have provided important insights into genome architecture (using imaging and chromosome capture methods)[3–5] as well as detailed genome-wide epigenetic profiles and expression maps (using various sequencing methods)[6–8]. However, obtaining high-throughput paired measurements of these different data modalities within single cells is still a major challenge requiring significant breakthroughs in single-cell technologies.

Different data modalities provide different perspectives on a population of cells and their integration is critical for studying cellular heterogeneity and its function (Fig. 1a). Current computational methods allow the integration of datasets of the same modality[9–11] or of different modalities with the same data structure such as various sequencing measurements[12–15]. We here present a computational framework based on autoencoders for integrating and translating between different data modalities with very distinct structures. Several works have proposed using autoencoders for domain adaptation (in particular batch correction) in the context of biological data[16,17]. Different from these works, our method uses autoencoders to integrate and translate between different data modalities that may have very different representations. A separate line of work has proposed using neural networks to directly translate between pairwise modalities in an unsupervised manner[18,19] or with side information[20,21]. These methods tend to focus on modalities with similar representations (e.g., image-to-image-translation) and directly translate between pairs of modalities without learning a common latent representation of the data. In contrast, our work maps each data distribution to a common latent distribution using an autoencoder. This not only enables data integration and translation between arbitrary modalities in a globally consistent manner, but, importantly, it also enables performing downstream analysis such as clustering across multiple modalities at once. Other work has proposed coupled autoencoders to translate between paired biological data[22], which differs from our method that does not require paired data. Building on Makhzani et al.[23], we align the latent space of an autoencoder using adversarial training and leverage this technique for data integration and/or translation. In particular, our framework can be applied to integrate and translate imaging and sequencing data, which cannot yet be obtained experimentally in the same cell, thereby providing a methodology for hypothesis generation to predict the genome-wide expression profile of a particular cell given its chromatin organization and vice-versa. Such a methodology is valuable to understand how features in one dataset translate to features in the other.

## Results

### Cross-modal autoencoders: Multi-domain data integration and translation using autoencoders.

To integrate and translate between data modalities with very distinct structures, we propose a new strategy of mapping each dataset to a shared latent representation of the cells (Fig. 1a, b). This mapping is achieved using autoencoders[24–26], neural networks consisting of an encoder (mapping to the latent space) and a decoder (mapping back to the original space), whose architectures can be customized to the specific data modality (Fig. 1b, c). Combining the encoder and decoder modules of different autoencoders enables translating between different data modalities at the single-cell level (Fig. 1d). To enforce proper alignment of the embeddings obtained by the different autoencoders, we employ a discriminative objective function to ensure that the data distributions from the different modalities are matched in the latent space. When prior

knowledge is available, an additional term in the objective function can be used that encourages the alignment between specific markers or the anchoring of certain cells. In the following, we formally introduce our framework.

We formalize the multi-modal data integration problem within a probabilistic framework. Each modality or dataset presents a different view of the same underlying population of cells. Formally, we consider cells from each modality $1 \leq i \leq K$ as samples of a random vector $X_i$ that are generated independently based on a common latent random vector $Z$:

$$X_i = f_i(Z, N_i), \quad \forall i = 1, \ldots, K, \tag{1}$$

where $f_i$ are deterministic functions, $Z$ has distribution $P_Z$, and $N_i$ are noise variables. The domain of $Z$, here denoted by $\mathcal{Z}$, represents some underlying latent representation space of cell state, and each function $f_i$ represents a map from cell state to data modality $i$. For simplicity of notation, we assume for the remainder of this section that each $X_i$ is 1-dimensional and obtained via a deterministic function of $Z$, so that the noise variables $N_i$ can be ignored. This model implies the following factorization of the joint distribution $P_{\mathbf{X}}$ (with density $p_{\mathbf{X}}$) of the data over all modalities:

$$p_{\mathbf{X}}(\mathbf{x}) = \int_{\mathcal{Z}} \Pi_{i=1}^K \, p_{X_i|Z}(x_i|z) p_Z(z) dz, \tag{2}$$

where $p_Z$ is the probability density of $Z$, and $p_{X_i|Z}$ is the conditional distribution of $X_i$ given $Z$ that reflects the generative process. Multi-modal data integration can then be formalized as the problem of learning conditional distributions $P_{X_i|Z}$ as well as the latent distribution $P_Z$ based on samples from the marginal distributions $P_{X_1}, P_{X_2}, \ldots P_{X_K}$, which are given by the datasets. Note that the assumption that each $X_i$ is obtained via a deterministic function of $Z$ implies that the latent distribution of each dataset is the same. However, by including the noise variables $N_i$ as in Equation (2), our method extends to the case where only a subset of the latent dimensions is shared between the different modalities and the remaining dimensions are specific to each modality.

When the latent distribution $P_Z$ is known, then learning the conditional distributions $P_{X_i|Z}$ given the marginals $P_{X_1}, P_{X_2}, \ldots, P_{X_K}$ can be solved by learning multiple autoencoders. Specifically, for each domain $1 \leq i \leq K$, we propose training a regularized encoder-decoder pair $(E_i, D_i)$ to minimize the loss

$$\mathbb{E}_{x \sim P_{X_i}}[L_1(x, D_i(E_i(x))) + \lambda L_2(E_i \# P_{X_i} | P_Z)], \tag{3}$$

where $\lambda > 0$ is a hyperparameter, $L_1$ is the (Euclidean) distance metric, $L_2$ represents a divergence between probability distributions, and $E_i \# P_{X_i}$ is the distribution of $X_i$ after embedding to the latent space $\mathcal{Z}$. Translation from domain $i$ to $j$ is accomplished by composing the encoder from the source domain with the decoder from the target domain, i.e.,

$$X_{i \to j}(x_i) := D_j(E_i(x_i)). \tag{4}$$

The autoencoders obtained by minimizing the loss in Equation (3) satisfy various consistency properties; see ref.[27].

Since $P_Z$ is not usually known in practice, it must also be estimated from the data. This can be done using the following approaches: (i) learn $P_Z$ by training a regularized autoencoder on data from a single representative domain; or (ii) alternate between training multiple autoencoders until they agree on an invariant latent distribution. The first approach is typically more stable in practice, while the second captures variability across multiple domains and is therefore more suitable for integrating multiple datasets. Note that $P_Z$ is by no means unique; there are multiple

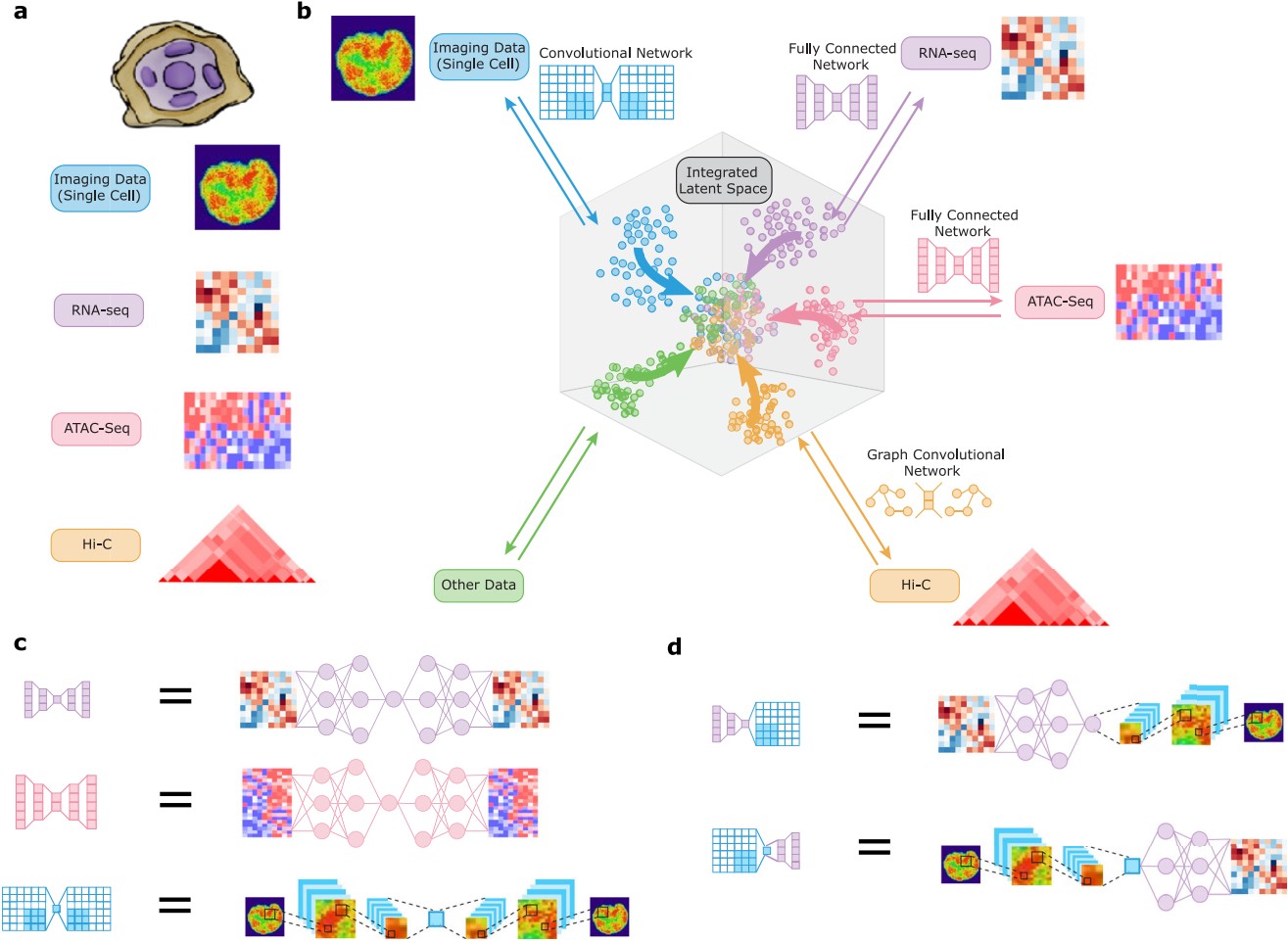

**Fig. 1 Schematic of multimodal data integration and translation strategy using our cross-modal autoencoder model. a** Each modality or dataset (represented by different colors) presents a different view of the same underlying population of cells of interest. **b** Our computational strategy to integrate multiple modalities involves embedding each dataset into a shared space that represents the latent state of the cells, such that the distributions of each dataset mapped into the latent space are aligned. **c** The embedding of each dataset is performed using an autoencoder, a neural network with separate encoder and decoder modules, whose architectures can be customized to the specific data modality (autoencoders for each modality are represented by different colors). **d** Combining the encoder and decoder modules of different autoencoders enables translation between different data modalities at the single-cell level.

solutions that can result in the same observed data distributions in the different domains.

To be concrete, an invariant latent distribution based on two domains $i, j \in \{1, \ldots, K\}$ is learned as follows. Let $\hat{P}_{Z_{i'}}$, $i' \in \{i, j\}$ denote the empirical latent distribution based on the encoded data from domain $i'$, i.e., $\hat{P}_{Z_{i'}} = E_{i'} \# P_{X_{i'}}$. Then for domain $i$, we optimize the objective

$$\min_{E_i, D_i} \mathbb{E}_{x \sim P_{X_i}} L_1(x, D_i \circ E_i(x)) + \lambda L_2(E_i \# P_{X_i} | P_{\hat{Z}_j}), \quad (5)$$

while for domain $j$, we optimize the objective

$$\min_{E_j, D_j} \mathbb{E}_{x \sim P_{X_j}} L_1(x, D_j \circ E_j(x)) + \lambda L_2(E_j \# P_{X_j} | P_{\hat{Z}_i}). \quad (6)$$

In practice, we parameterize $(E_i, D_i)$ by neural networks and minimize the objective function via stochastic gradient updates. In particular, $L_2$ can be chosen to be the discriminative loss,

$$L_2(P|Q) := \max_f \mathbb{E}_{x \sim P} \log f(x) + \mathbb{E}_{x \sim Q} \log (1 - f(x)), \quad (7)$$

which is equivalent to the Jensen-Shannon divergence up to a constant factor. In practice, the model architecture of each autoencoder is selected based on the input data representation

(e.g., fully-connected network for gene expression data and convolutional network for images). The dimensionality of the latent distribution is a hyperparameter that is tuned to ensure that the autoencoders are able to reconstruct the respective data modalities well. For sequencing data, PCA can be used to obtain an initial estimate of the intrinsic dimensionality of the data, which can then be fine-tuned by analyzing the reconstruction loss of the model. For imaging data the reconstruction quality can also be assessed qualitatively (see Supplementary Fig. 5) and a variational autoencoder with a small weight on the KL divergence regularization term can be used to improve image generation quality.

**Incorporating prior knowledge.** Prior knowledge is sometimes available to guide the integration of different data modalities. For example, there may be knowledge of alignment of specific markers or clusters, or knowledge of certain samples from different datasets corresponding to the same cell, i.e., the same point in the latent space. In this case, training of the autoencoders can be guided by additional loss functions that incorporate the prior knowledge.

Discriminative loss to align shared markers/clusters among datasets: If there are shared markers or clusters that are present in

two datasets, they can be aligned by replacing $L_2$ above with the following discriminative loss that is conditioned on these factors:

$$L_2(P|Q) := \max_f \mathbb{E}_{x,y\sim P} \log f(x,y) + \mathbb{E}_{x,y\sim Q} \log(1 - f(x,y)),$$

$$(8)$$

where $P$ and $Q$ are now joint distributions over the data and the markers and/or clusters. This approach is valid for both discrete and continuous values of the cluster/marker $y$. For example, in ref. [27], this approach was used to align a continuous differentiation marker between RNA-seq and ChIP-seq data. Alternatively, if the markers or clusters can take $m$ discrete values (i.e., 1, …, $m$), then we can add a simple classifier model $p_\theta(Y|Z)$ with parameters $\theta$ and minimize the loss

$$\sum_{\text{modality } i} \mathbb{E}_{x,y\sim P_i} \sum_{j=1}^{m} \mathbf{1}(y = j) \, p_\theta(Y = j|Z = E_i(x)) \qquad (9)$$

with respect to $\theta$ and the parameters of the encoders $E_i$; here $P_i$ is the distribution of the $i$th data modality. This loss function encourages data points with the same class label irrespective of the data modality to be clustered together in the latent space.

Anchor loss to match paired samples: If $(x_1, x_1')$, $(x_2, x_2')$, … , $(x_m, x_m')$ are corresponding points from two datasets that are embedded by encoders $E$ and $E'$, we can add the following anchor loss,

$$\sum_{i=1}^{m} ||E(x_i) - E'(x_i')|| \qquad (10)$$

to minimize their distance in the latent embedding space.

**Model validation on paired single-cell RNA-seq and ATAC-seq data.** Recent technological advances have made it possible to obtain paired single-cell RNA-seq and ATAC-seq data. Such paired data was collected from human lung adenocarcinoma-derived A549 cells treated with dexamethasone (DEX) for 0, 1, or 3 hours in ref. [28]. While our autoencoder framework is designed to integrate vastly different data structures, in the following we show that our framework is competitive with previous methods for the simpler problem of integrating different modalities with similar data structures. For details on the implementation see Supplementary Methods, Supplementary Table 1, Supplementary Table 2, and Supplementary Data 1. Since the RNA-seq and ATAC-seq data was collected in the same cell, we could evaluate the accuracy of our method in matching samples from RNA-seq to ATAC-seq (and vice-versa). We evaluated the accuracy of the matching by the following two measures: (a) the fraction of cells whose cluster assignment (0, 1, or 3 hours treatment with DEX) is predicted correctly based on the latent space embedding, and (b) $k$-nearest neighbors accuracy, i.e., the proportion of cells whose true match is within the $k$ closest samples in the latent space (in $\ell_1$-distance) or in the original space for methods that do not rely on the latent space.

In Fig. 2, we compare our cross-modal autoencoder model to methods that align modalities in the latent space, namely deep canonical correlation analysis (DCCA)[29], which determines a nonlinear transformation of the two datasets to maximize the correlation of the resulting representations, as well as Seurat, a prominent method for biological data intergration of similar modalities[9,12]. In addition, we compare our cross-modal autoencoder model to two additional methods that do not rely on the latent space for alignment of modalities, namely CycleGAN[19] and MAGAN[21]. Similar to CycleGAN, our cross-modal autoencoder does not require paired samples, which is advantageous for many modalities, where the process of data collection results in destruction of the cell (e.g., RNA-seq) and thus the same cell

cannot be used in another assay to measure a different modality (e.g., imaging). However, if additional information is available such as shared markers measured in all modalities and/or paired data, similar to the MAGAN approach, this prior information can be incorporated through additional terms in the loss function (see section on incorporating prior knowledge). In terms of comparisons with methods that align modalities in the latent space, our autoencoder framework outperforms Seurat and is competitive with DCCA for integrating single-cell RNA-seq and single-cell ATAC-seq data both in terms of fraction of cells assigned to the correct cluster (Fig. 2a) as well as $k$-nearest neighbor accuracy (Fig. 2b). While paired data was only used to evaluate the accuracy in Figs. 2a, b and 2c–e explore the setting in which paired data on a fraction of samples is used for training. Although paired data is not necessary for our method, such prior knowledge can be incorporated using the anchor loss described above, which ensures that paired samples are close in the latent space. Figure 2c, d show that our autoencoder model outperforms DCCA, CycleGAN and MAGAN when trained on varying amounts of paired data. In fact, as shown in Fig. 2e, our autoencoder model trained with just 25% of the paired samples has similar performance to DCCA trained on all (i.e., 100%) of the paired samples, thereby indicating that our method is practical and competitive also in the setting where some paired data is available.

**Experimental validation on single-cell RNA-seq and chromatin images of naive CD4+ T-cells.** We applied our method to integrate single-cell RNA-seq data with chromatin images in order to study the heterogeneity within naive T-cells. T-cell activation is a fundamental biological process and identifying naive T-cells poised for activation is critical to understanding immune response[30]. Moreover, linking genome organization with gene expression generates hypotheses that can be tested experimentally to validate our methodology.

**Single-cell RNA-seq analysis of naive CD4+ T-cells revealed two distinct subpopulations.** We analyzed single-cell RNA-seq data of human peripheral blood mononuclear cells (PBMCs) from ref. [31]; for details on the analysis see Supplementary Methods. We used known markers to identify naive and activated (CD4+) T-cells (Fig. 3a and Supplementary Fig. 1, Supplementary Table 3, Supplementary Data 2. An in-depth analysis of the naive T-cell population revealed two distinct subpopulations (Fig. 3a, see Methods). The number of subpopulations/clusters was obtained via two separate analyses, namely by maximizing the silhouette coefficient (Supplementary Fig. 2a) and by minimizing the Bayesian information criterion (Supplementary Fig. 2b). The co-association matrix shown in Fig. 3b, which quantifies how often each pair of cells was clustered together for different clustering methods, shows that the two clusters were highly robust to the choice of clustering method. Differential gene expression and gene ontology (GO) enrichment analysis indicated that one cluster corresponded to quiescent cells while the other was poised for activation, with an expression profile more similar to that of activated T-cells (Fig. 3c, d and Supplementary Data S2). Specifically, we observed that one of the two clusters of naive CD4+ T-cells contained "immune response" and "cell activation" as one of the top significant GO terms as well as a well-known activation marker IL32 as one of the differentially expressed (DE) genes.

**Analysis of single-cell chromatin images of naive CD4+ T-cells revealed two distinct subpopulations.** Given the link between expression and chromatin organization[32], we hypothesized the presence of two subpopulations of naive T-cells with distinct

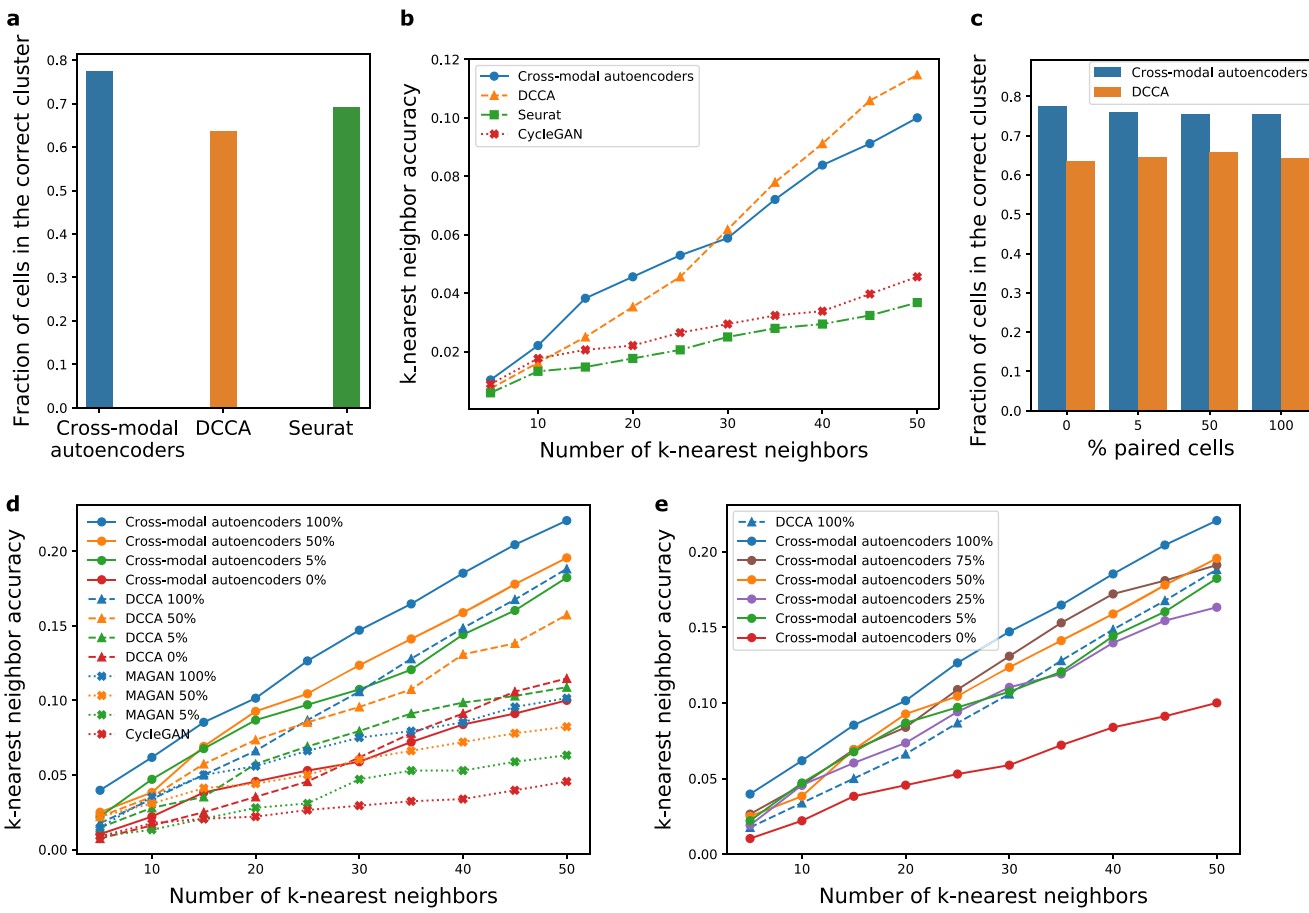

**Fig. 2 Performance of our multimodal data integration method (cross-modal autoencoders), deep canonical correlation analysis (DCCA), Seurat, CycleGAN, and MAGAN on paired RNA-seq and ATAC-seq data. a** Fraction of cells that were assigned to the correct treatment time cluster based on their embedding in the integrated latent space that was learned by fitting our cross-modal autoencoder model, DCCA, or Seurat. **b** $k$-nearest neighbor accuracy for quantifying the quality of matching between local neighborhoods for our cross-modal autoencoder model, DCCA, Seurat, and CycleGAN trained with 0% supervision (no paired samples). **c** Fraction of cells that were assigned to the correct treatment time cluster for our cross-modal autoencoders and DCCA trained with varying amount of paired samples. **d** $k$-nearest neighbor accuracy for our cross-modal autoencoders, DCCA, MAGAN, and CycleGAN trained with 0, 5, 50, and 100% of the paired samples. **e** $k$-nearest neighbor accuracy for our cross-modal autoencoder model trained with varying amount of paired samples versus DCCA trained on all paired samples. In **a–c** colors denote different domain translation methods and in **d–e** colors denote different levels of supervision (paired samples). Additionally, different markers denote different domain translation methods.

chromatin packing features. To test this, we carried out DAPI-stained imaging experiments of naive CD4+ human T-cells and analyzed their chromatin organization (Methods, Fig. 3e, and Supplementary Fig. 3). We extracted image features by quantifying the chromatin density in concentric spheres with increasing radii (Methods, Fig. 3f). Cluster analysis based on the extracted features revealed two distinct subpopulations of cells, with higher chromatin density in the central and peripheral nuclear regions respectively (Fig. 3g, Supplementary Fig. 4). These observations are consistent with previous experiments in mouse naive T-cells that also showed two subpopulations with distinct chromatin organization patterns, where naive T-cells with more central heterochromatin were shown to be poised for activation[33].

**Cross-modal autoencoder framework allows integrating and translating between single-cell expression and imaging data.** Up to this point, we had observed two subpopulations of naive T-cells based on a separate analysis of gene expression (from single-cell RNA-seq data) and chromatin packing (from single-cell imaging data). To link the identified subpopulations from the unpaired datasets, we used our cross-modal autoencoder framework to integrate the single-cell RNA-seq data with the

chromatin images (Methods and Supplementary Table 4), thereby enabling translation between the two data modalities at the single-cell level (Fig. 4a and Supplementary Fig. 5). Visual inspection of the latent representations indicates that the subpopulations from the two datasets are appropriately matched (Fig. 4b and Supplementary Fig. 6). To quantitatively assess whether our methodology aligns imaging features and gene expression features in a consistent manner, we next analyzed the latent embeddings as well as the results of translation between the two datasets. Consistent with other methods used for data integration and translation in the biological domain, where the goal is to provide a matching between samples in the observed datasets[12], our evaluation is based on the full dataset used for training rather than a held-out evaluation set.

**ROC analysis on translated datasets indicates that imaging and gene expression features are consistently aligned.** In order to assess whether translated image (or RNA-seq respectively) datasets are still able to separate poised and quiescent subpopulations (or central and peripheral subpopulations respectively) and analyze if the clusters obtained separately from gene expression and imaging datasets align with each other, we performed Receiver

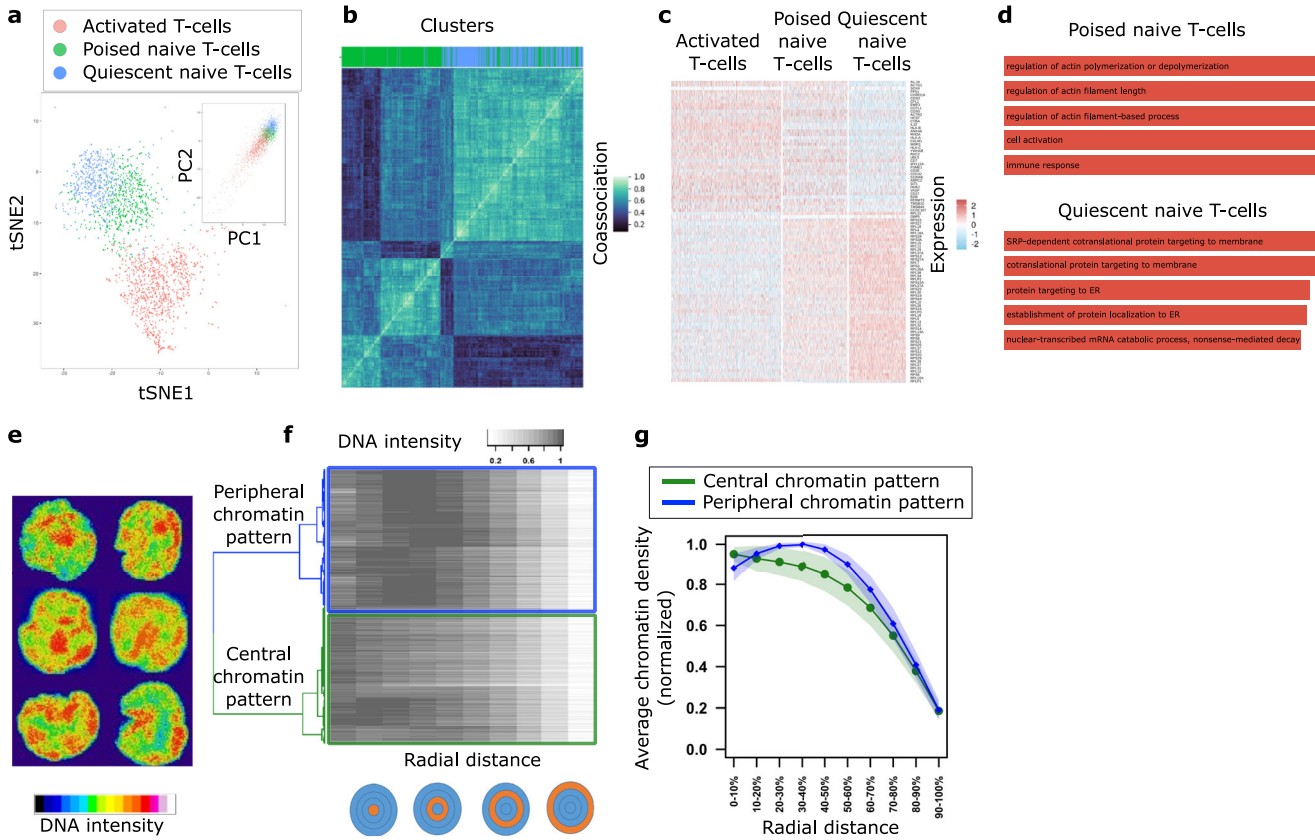

**Fig. 3 Analysis of single-cell RNA-seq data and single-cell chromatin images of naive CD4+ T-cells reveals two distinct subpopulations respectively.**
**a** t-SNE and PCA (inset) embeddings of single-cell RNA-seq data derived from[31]. Cluster analysis reveals activated (red) population of T-cells and naive population of T-cells divided into two subpopulations (poised and quiescent, denoted in green and blue, respectively). **b** Consensus clustering plot demonstrating the robustness of quiescent (blue) and poised (green) clusters of naive T-cells to various clustering methods. Gene expression data was clustered using k-means, Gaussian mixture models, and spectral clustering based on a k-nearest neighbor graph with $k \in \{10, 20, 50, 100\}$ with 100 initializations for each method. **c** Differential gene expression analysis between the blue and green subpopulations reveals two distinct gene expression programs. The green subpopulation of naive T-cells is more similar to the activated T-cells and hence poised for activation, while the blue subpopulation shows an upregulation of ribosomal genes and has a relatively more quiescent expression profile. **d** Gene ontology enrichment analysis of marker genes for quiescent and poised naive T-cell subpopulations supports two distinct gene expression programs. **e** Examples of DAPI-stained nuclear images of naive CD4+ T-cells. **f** Cluster analysis of the 3D nuclear images is performed by first quantifying the chromatin signal in concentric spheres with increasing radii, and then using hierarchical clustering on these spatial chromatin features. The features were clustered using hierarchical clustering with complete linkage based on the distance matrix obtained from 1-Spearman's correlation. **g** Average chromatin signal, calculated using $n = 729$ cells from two biologically independent replicates, (mean represented by the solid line and standard deviation represented by shading) in concentric spheres with increasing radii for central (green) and peripheral (blue) clusters. One cluster has higher concentration of chromatin in the central region of the nucleus (green), while the other cluster has higher concentration of chromatin in the peripheral region of the nucleus (blue).

Operating Characteristic (ROC) analysis on the translated datasets. For RNA-seq, we first trained a random forest classifier (using 100 trees in a forest with 2 as the maximum depth of a tree) on the RNA-seq data with labels based on poised versus quiescent clustering of naive CD4+ T-cell gene expression data. This classifier learned the genes that separate the two clusters. Next, we translated chromatin images into RNA-seq using our autoencoder method and assessed the performance of the pre-trained classifier on its ability to separate central versus peripheral clusters on images translated to RNA-seq (Fig. 4c, top). Similarly, to assess translation of RNA-seq into images, we trained a classifier to separate central versus peripheral chromatin patterns. Then, we translated RNA-seq data into images and evaluated the performance of the pre-trained classifier in being able to separate poised versus quiescent clusters (Fig. 4c, bottom). The area under the curve (AUC) was computed for both of these tasks. The high AUCs demonstrate that classifiers trained to distinguish between the subpopulations in the

original datasets also performed well when evaluated on the translated datasets.

**Strong correlation of DE genes between original RNA-seq and images translated to RNA-seq indicates consistent alignment.** Imaging datasets can provide a rich quantification of cells, such as their chromatin organization. Based on image analysis, subpopulations of cells with different characteristics may be found (e.g., central versus peripheral chromatin organization), and it is often of interest to study which genes might be markers of each subpopulation such that these subpopulations can be separated for example using antibodies against the marker genes. However, generally the full gene expression and imaging features cannot be measured in the same cell. Our computational framework can translate chromatin images into RNA-seq and calculate the predicted mean difference in expression between the subpopulations (e.g., for central versus peripheral chromatin organization). As shown in Fig. 4d, the observed mean

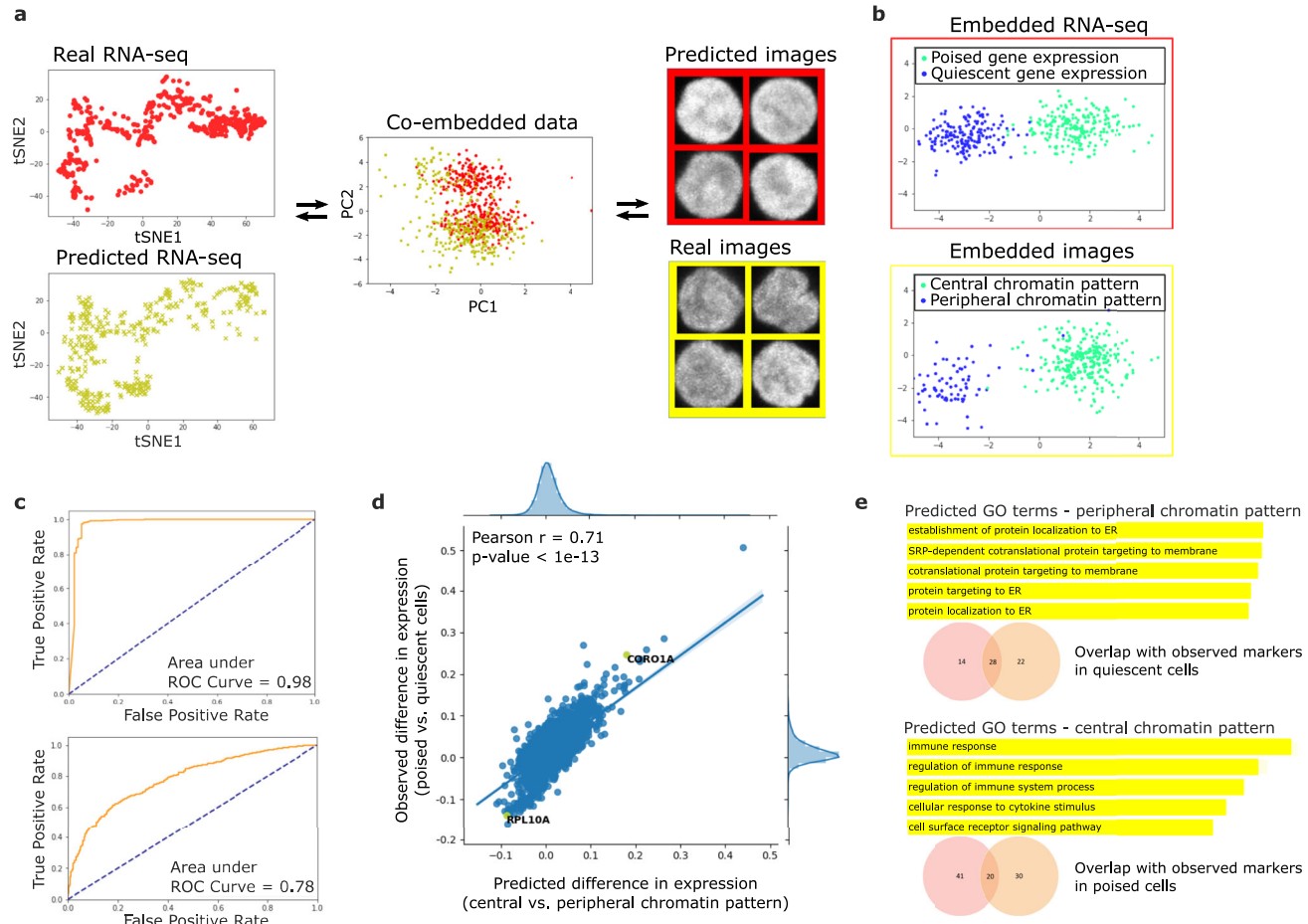

**Fig. 4 Integration of single-cell RNA-seq data and single-cell nuclear images of naive T-cells using our methodology allows translating between chromatin packing and gene expression profiles. a** Illustration of data integration and translation: (left) t-SNE plots of observed single-cell RNA-seq data (red) and single-cell RNA-seq data translated from single-cell images (yellow); (middle) PCA visualization of single-cell RNA-seq data (red) and single-cell imaging data (yellow) embedded in 128-dimensional latent space; (right) examples of observed single-cell images (yellow) and images translated from single-cell RNA-seq data (red). **b–e** Evidence that our data integration methodology correctly aligns gene expression features and imaging features. **b** Linear Discriminant Analysis (LDA) plots of single-cell RNA-seq (top) and imaging (bottom) datasets embedded in the latent space. The clusters with more quiescent (blue) and poised (green) gene expression programs from the RNA-seq dataset are aligned with the clusters with peripheral (blue) and central (green) chromatin patterns from the imaging dataset. **c** (top) Receiver Operating Characteristic (ROC) curve illustrating performance of a classifier trained to distinguish between peripheral and central chromatin patterns in images when evaluated on images translated from RNA-seq data. (bottom) ROC curve illustrating performance of a classifier trained to distinguish between quiescent and poised gene expression programs when evaluated on RNA-seq data translated from images. High performance of both classifiers indicates that the alignment of the clusters in the latent space in **b** also holds in the original gene expression and imaging spaces. The dotted line represents random guessing based on evenly-distributed classes. **d** Differential gene expression analysis between cells with central and peripheral chromatin pattern performed on the predicted gene expression matrix translated from images using our methodology. The predicted fold-change of gene expression based on images is strongly correlated with the observed fold-change of gene expression between quiescent and poised naive T-cells from the actual RNA-seq dataset. **e** Analysis of gene ontology (GO) enrichment terms of cells with central and peripheral chromatin pattern based on the predicted gene expression matrix translated from images using our methodology shows a high overlap between predicted markers (orange) from the imaging dataset and actual markers (red) from the RNA-seq dataset.

difference in expression is strongly correlated with the predicted mean expression difference. In addition, we obtained a set of marker genes associated with central and peripheral chromatin organization by performing two-sided Welch's *t*-test on the generated RNA-seq data (considering marker genes for each cluster to be the top 50 genes that had the highest mean difference in expression for the two clusters as well as *p*-value < 0.05 after adjustment for multiple hypothesis testing using the Benjamini-Hochberg procedure). Note the considerable overlap between the true and predicted marker genes (Fig. 4e). We also performed GO analysis on the marker genes for each cluster; we report the top 5 GO biological process terms with lowest *p*-values (FDR adjusted *p*-value < 0.05). In summary, in the gene expression matrix translated from the imaging dataset, we found

that the differential expression of genes was strongly correlated with the true observed differential gene expression and that the predicted and observed marker genes showed considerable overlap.

**Experimental validation of matching via protein immuno-fluorescence staining.** Our model generates predictions of gene expression programs based on patterns of chromatin density (Fig. 4e). To validate these results experimentally, we chose two genes, *CORO1A* and *RPL10A*, which were predicted to be strongly upregulated in the naive T-cell subpopulations with central and peripheral patterns of chromatin density respectively (Figs. 4d, 5a). We analyzed the immunofluorescence staining data of these proteins obtained along with chromatin images (Fig. 5b). Consistent with the

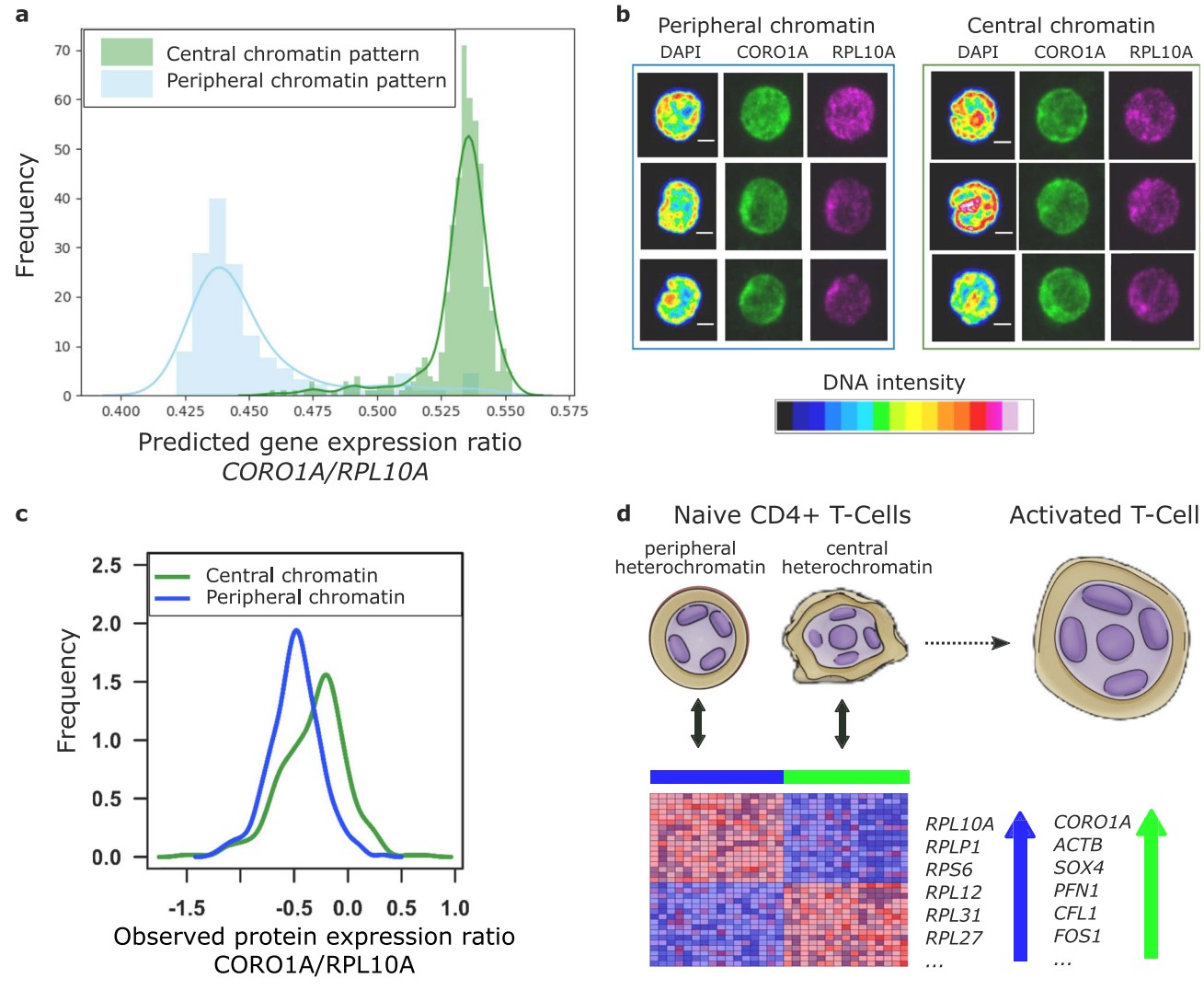

**Fig. 5 Validation of our model alignment using single-cell immunofluorescence experiments. a** Histograms of predicted *CORO1A/RPL10A* gene expression ratio in cells with central (green) and peripheral (blue) chromatin pattern based on the gene expression matrix translated from the imaging dataset. Our model predicts the upregulation of *CORO1A* and *RPL10A* in the cells with central and peripheral chromatin patterns respectively. **b** Examples of immunofluorescence staining data of CORO1A and RPL10A proteins collected along with the chromatin images. **c** Histograms of measured CORO1A/RPL10A protein ratio in cells with central (green) and peripheral (blue) chromatin pattern. Consistent with the model prediction, CORO1A and RPL10A proteins are upregulated in the cells with central and peripheral chromatin patterns respectively ($p$-value $< 2.2 \times 10^{-16}$, two-sided Welch's $t$-test). **d** Schematic of the two naive T-cell subpopulations characterized by our multimodal analysis, in which peripheral and central patterns of chromatin density are associated with gene expression programs for quiescent (blue) and poised (green) naive CD4+ T-cells respectively. The up and down arrows represent which genes are upregulated and downregulated respectively as predicted by our model.

model predictions, we found that CORO1A was upregulated in the cells with central chromatin pattern, while RPL10A was upregulated in the images with peripheral chromatin pattern (Fig. 5c and Supplementary Methods and Supplementary Fig. 7). These results altogether demonstrate that our method properly aligns the gene expression and image features that characterize two distinct subpopulations of human naive T-cells, and suggests that peripheral and central enrichment of chromatin are associated with gene expression programs for more quiescent and poised naive CD4+ T-cells respectively (Fig. 5d).

## Discussion

In summary, we presented a powerful approach to integrate and translate between different data modalities of very different structures, namely single-cell chromatin images and RNA-seq. Using our cross-modal autoencoder methodology, we established a quantitative link between chromatin organization and

expression, jointly characterizing a subpopulation of naive T-cells that is poised for activation using both data modalities. Additionally, we validated our model's predictions of gene expression using protein fluorescence experiments.

While we used our method to align RNA-seq and imaging datasets, we have presented a general framework that can be adapted to numerous other biological problems. As indicated in Fig. 1, our framework can be used to integrate datasets of different modalities simply by incorporating autoencoder architectures tailored to those modalities. For example, Hi-C data could be integrated using a graph neural network and multi-channel cell images using a convolutional neural network with different input channels. Also, while we focused on aligning datasets each containing two distinct clusters, our method can be applied to datasets with other distributions as long as the samples are taken from the same cell population. For example, in applications where there are no clear clusters in the datasets, our method can be used to align continuous

markers between datasets by conditioning the adversarial loss on the values of the continuous marker (Equation 3). In applications where there is some shared signal between modalities as well as a signal that is individual to each modality, our model can be extended by introducing a subset of latent dimensions that is specific to each modality. Empirically validating this aspect of our model is a potential direction for future work. An important consideration, however, is that while our method can be applied for data integration and cross-modal alignment in generic contexts, depending on the data distributions, there may be multiple alignments that satisfy the same objective function. Additional constraints (in the form of prior knowledge) should be added to these models where possible to enforce alignments that are biologically accurate. Overall, we envision an iterative process of biological discovery where our predictive model is used for hypothesis generation (for example linking particular image features to particular gene regulatory modules), the hypotheses are validated (or disproved) experimentally, and the new experimental results now serve as additional data (prior knowledge) for improving the alignment of the model. In summary, our methodology can be applied generally to integrate single-cell datasets that cannot yet be measured in the same cell by using a different autoencoder for each data modality, and as such has broad implications for the integration of spatial transcriptomics[34], proteomics[35] and metabolomics[36] datasets. In particular, our methodology can be applied to generate hypotheses and predict the functional landscape of single cells in a tissue section where only limited functional data is available by acquiring chromatin imaging data.

## Methods

**Cell culture and immunostaning**. CD4+/CD45RA+ naive helper T-cells from human peripheral blood were purchased from AllCells. These cells were revived and cultured in media (RPMI-1640 + 10% FBS + 1% pen-strep) as per the manufacturer's instructions. The cells for the experiments were used within two days upon revival.

Cells in media were allowed to adhere to Poly-lysine coated slides for 30 minutes. Cells were then fixed with 4% Paraformaldehyde (Sigma) for 30 minutes and washed with PBS three times, which also removed unattached cells. Permeabilization was done with 0.5% Triton X-100 (Sigma) for 10 minutes followed by PBS washes. Blocking was done with 5% BSA in PBS for 30 minutes and incubated with primary and secondary antibodies as per the dilution and incubation time recommended by the manufacturer. The primary antibodies used in this study are anti-RPL10A antibody (Abcam, ab174318, dilution 1/200) and Anti-Coronin 1a/TACO antibody (Abcam ab14787, dilution 1/150). Cells were washed with PBS (+0.1% Tween) three times after primary and secondary antibody incubation. During the final step, excess liquid was removed by slanting the slides. ProLong® Gold Antifade Mountant with DAPI (ThermoFischer Scientific) was added to these slides and allowed to cure for 24 hours. Coverslips were then sealed and imaged using a confocal microscope.

**Confocal microscopy and image analysis**. 1024 × 1024 and 12-bit multi-channel images were obtained using a Nikon A1R confocal microscope. Z-stack images were captured using a 100× objective with a pixel size of 0.1 $\mu m$ and 0.5 $\mu m$ depth. Images were processed and further analyzed using custom programs in Fiji and R (see below in code availability).

The nuclear boundaries were segmented in 3D using the DAPI channels to identify individual nuclei. These nuclei were eroded by 0.5 microns in $x$, $y$, and $z$ iteratively until the volume of the eroded nucleus was less than 10 cubic microns. Then the mean intensity of each 3D ring (width 0.5 microns) in the nucleus was computed for all cells. The intensity fraction was calculated by normalizing the mean ring intensity for each nucleus (maximum = 1). Linear interpolation was then used to compute the intensity fraction of rings that occupy 0–10% to 90–100% volume fraction of the nucleus. The heatmaps were visualized using functions from gplots, RColorBrewer and dendextend.

In order to calculate the cellular levels of proteins, the 3D nuclear object was dilated by 2 microns in $x$, $y$, and $z$. This was efficient as the cells were all spherically shaped with high karyoplasmic index. The total intensity in the 3D cellular object was computed for each protein channel and their ratio was obtained for each cell.

**Gene expression analysis of naive CD4+ T-cells**. We aimed to explore potential heterogeneity in naive CD4+ T-cell gene expression in relation to CD4+ T-cells that already underwent activation. We performed a feature selection step, keeping

genes which had average log-fold change of >0.05 between naive and activated CD4+ T-cells (and vice-versa), resulting in 1187 genes. Similar to the analysis of PBMCs (see Supplementary Methods), we applied PCA for dimensionality reduction on the selected genes, keeping the top 30 components and clustered the naive CD4+ T-cells using the default clustering method in Seurat version 2.3.0 with resolution of 0.8 (Supplementary Fig. 1c). Based on differential expression analysis and t-SNE embedding, the smallest cluster (shown in gray in Supplementary Fig. 1c) was determined to belong to the CD8+ T-cell population since the top differentially overexpressed genes for this small cluster were CD8A and CD8B. Therefore, this small cluster was removed from the downstream gene expression analysis of the naive CD4+ T-cells. In order to characterize the remaining two subpopulations, we performed differential expression analysis on the two subpopulations of naive CD4+ T-cells using Wilcoxon rank sum test. We defined marker genes as all genes with Bonferroni-corrected $p$-value of <0.05. Fig. 3c, shows the resulting heatmap for the genes that are markers between poised and quiescent subpopulations of naive T-cells and are also part of the 1187 genes that have an average log-fold change of >0.05 between naive and activated CD4+ T-cells (and vice-versa). Gene ontology analysis was performed on these marker genes overexpressed in each cluster (average log-fold change > 0) using g:Profiler[37,38], keeping the top 5 gene ontology biological process terms with lowest $p$-values (Fig. 3d). All reported $p$-values (after adjusting for multiple hypothesis testing using the Benjamini-Hochberg procedure) were ≤0.05.

Since the identification of the two subpopulations of naive T-cells is an important step in our analysis, we thoroughly evaluated its robustness with respect to number of clusters and clustering methods. We re-clustered the data corresponding to naive CD4+ T-cells using Seurat version 2.3.0 with different resolution parameters, i.e., 0.9, 1.1, and 1.15 to obtain 3, 4, and 5 clusters respectively. We computed the silhouette coefficient for each clustering, observing that the number of clusters corresponding to 2 gave the highest score (Supplementary Fig. 2a). This suggests that using 2 clusters is optimal. We also fit a Gaussian mixture model to the data and computed the Bayesian information criterion (BIC) for a model with 1, 2, 3, 4, and 5 mixture components (across 100 randomly initialized trials). Also with this method the model with 2 components resulted in the lowest mean BIC, suggesting again that 2 clusters is optimal for this data (Supplementary Fig. 2b). To test the robustness with respect to different clustering methodologies, we also used k-means, Gaussian mixture models, and spectral clustering based on a $k$-nearest neighbor graph with $k \in \{10, 20, 50, 100\}$ to cluster the data. We performed 100 different initializations for each method and computed the co-association matrix, which quantifies how often each pair of cells was clustered together; the result is shown in Fig. 3b. We observe that the chosen clustering given by Seurat is in strong agreement with the other methods and that the clusters are highly robust to the choice of the clustering method.

**Autoencoder training for integration and translation between single-cell RNA-seq data and single-cell chromatin images**. Images were normalized to range between [0, 1] and RNA-seq matrix was $\log(x + 1)$ normalized. Since the imaging dataset is more difficult to reconstruct in comparison to the RNA-seq dataset, we first pretrained the image autoencoder to reconstruct single-cell chromatin images for 850 epochs using the reconstruction loss and the discriminative loss in Equation (9). Subsequently, we trained the full model consisting of the pretrained image autoencoder, the RNA-seq autoencoder, and latent space discriminator using reconstruction loss and discriminative loss with hyperparameters $\lambda_1 = 0.1$, $\lambda_2 = 1$. The architectures of all networks are shown in Supplementary Table 4. Models were trained with the Adam optimizer with a learning rate of 1e-3. In Supplementary Figs. 8–9, we show that our findings are robust to the choice of architecture (fully-connected versus convolutional layers, number of layers, as well as latent space dimension).

**Supplementary materials**. Supplementary Methods, Supplementary Figs. 1 to 8, Supplementary Tables 1 to 4, Supplementary Data 1, Supplementary Data 2.

**Reporting summary**. Further information on research design is available in the Nature Research Reporting Summary linked to this article.

## Data availability

The data for model validation on paired single-cell RNA-seq and ATAC-seq is publicly available and was obtained from GSE117089[28]. The RNA-seq data for integration of RNA-seq and chromatin images is publicly available and was obtained from https://support.10xgenomics.com/single-cell-gene-expression/datasets/2.1.0/pbmc8k. The chromatin images are available at Zenodo from https://doi.org/10.5281/zenodo.4265737.

## Code availability

The code for model training is available at[39]: https://github.com/uhlerlab/cross-modal-autoencoders. Code containing the image processing scripts for the analysis of the primary images is available at[40]: http://github.com/SaradhaVenkatachalapathy/Radial_chromatin_packing_immune_cells. Data analysis was performed using standard libraries and software such as scikit-learn, scipy, numpy, seaborn and R.

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

## Acknowledgements

We thank Diego Pitta de Araujo for the schematic drawings. K.D.Y. was supported by the National Science Foundation (NSF) Graduate Research Fellowship and ONR (N00014-18-1-2765). A.B. was supported by J-WAFS and J-Clinic for Machine Learning and Health at MIT. The G.V.S. laboratory thanks the Mechanobiology Institute (MBI), National University of Singapore (NUS), Singapore and the Ministry of Education (MOE) Tier-3 Grant Program for funding. A.R. was supported by the National Science Foundation (DMS-1651995). C.U. was partially supported by NSF (DMS-1651995), ONR (N00014-17-1-2147 and N00014-18-1-2765), a Sloan Fellowship, and a Simons Investigator Award. The Titan Xp used for this research was donated by the NVIDIA Corporation.

## Author contributions

K.D.Y., A.B., S.V., A.R., G.V.S., and C.U. designed the research. K.D.Y., A.B., S.V., A.K., and A.R. performed model and data analysis. S.V. and K.D. performed experiments and analysis. K.D.Y., A.B., S.V., A.R., G.V.S., and C.U. wrote the paper.

## Competing interests

The authors declare no competing interests.
