## [Peer Review File · Nature Communications]

REVIEWER COMMENTS

Reviewer #1 (Remarks to the Author):

This paper describes an autoencoder for domain adaptation, which is also similar to the problem of batch correction. Both of these problems require an alignment of datasets, and has been characterized by many papers as manifold alignment. Moreover, autoencoders [7], and VAEs [6] have been used for this problem several times. Moreover, there is a whole set of works using cycleGANs [7,3] for both cross modal data alignment. In addition, there are approaches that use diffusion geometry and manifold learning methods for this purpose [7] showing results on single cell RNA-seq and ATAC-seq like this manuscript. However, the authors seem completely unaware of this body of literature, do not cite a single one of these papers, and don't compare the methods.

[1] Liu R, Zou J, and Balsubramani A. Learning Transport Cost From Subset Correspondence. ICLR, 2020.

[2] Makhzani A, Shlens J, Jaitly N, Goodfellow I, Frey B. Adversarial Autoencoders. arXiv:151105644 [cs]. May 2016.

[3] Amodio M, Krishnaswamy S. MAGAN : Aligning Biological Manifolds. ICML 2018.

[4] Amodio M, Krishnaswamy S. Image-To-Image Translation by Transformation Vector Learning CVPR 2019

[5] Lopez et al. Deep generative models for single-cell transcriptomics, Nature Methods 2018

[6] Stanley et al. Harmonic Alignment, SIAM Data Mining 2020

[7] Amodio et al. Exploring single-cell data with deep multitasking neural networks. Nature Methods 2019

[8]Zhu et al. Unpaired Image-to-Image Translation using Cycle-Consistent Adversarial Networks ICCV 2017

For all of these methods the authors need to formulate comparisons, put their work in context and describe conceptual differences against. Moreover the authors need to relate their adversarial setup to an adversarial autoencoder [2]

One other point to consider is that the densities may be different in the different modalities. Alignment of geometries over densities is considered in [6].

The only point of impact may be the exact modes that they are aligning, spatial data with non-spatial sequencing data. Spatial sequencing measurements does require convolutional layers which also some of the above references use especially [8,4], but image feature extraction on single cell data may be different than natural image feature extraction. This is the point which needs to be explored in great detail which this manuscript does not do.

Reviewer #2 (Remarks to the Author):

Yang et al describe an exciting and novel approach to integrate single-cell imaging and gene expression data. The primary innovation is the autoencoder training procedure and the new loss functions to incorporate markers shared among datasets (discriminative loss) and paired sample groups (anchor loss). The authors first benchmark their model in a paired gene expression and ATACseq dataset. They observe that their approach outperforms deep canonical correlation analysis based on two evaluation metrics. Next, the authors apply their approach to a dataset with gene expression and DAPI image measurements. The authors demonstrate, through a series of anecdotes based on independent clustering, that DAPI images and gene expression data share common signal. Overall, the manuscript is very interesting and timely, but I have several major and minor concerns that if addressed, will improve the manuscript.

Major Concerns

- There are several reporting and methodology concerns that I elaborate on in the minor concerns below. Individually, each of these concerns are mostly minor, but taken together, the many incomplete methods and reporting causes a major concern.
- The authors need to expand their discussion. An emphasis on limitations of their approach should be a primary focus. In the main experiment, the authors used a nice dataset that appears to have clean separations between two groups in both imaging and expression data modalities. What happens to their approach in the much more common scenario where there are no clear groupings or even if there are different numbers of groups between different data modalities? What about more complicated imaging assays beyond DAPI staining? How do the authors propose to use their method to make biological discoveries? Don't we expect certain signal to be unique to each modality? How can we use this approach to determine which signals are unique?
- The authors include a training and testing set in the ATACseq + gene expression application. It is not clear if the authors use a holdout test set in the image-expression translation evaluation. The authors note "We held out 10% of the data as test data and trained until the reconstruction loss on the test data was smaller than 10⁻³". This is not a true holdout set. Also, the authors do not note if their evaluations are based on the full dataset used for training.

Minor Concerns

- Some sentences convey more confidence than the results warrant. For example, a sentence in the second paragraph of the introduction states: "In particular, our framework can be applied to integrate and translate imaging and sequencing data, which cannot yet be obtained experimentally in the same cell, thereby providing a methodology to predict the genome-wide expression profile of a particular cell given its chromatin organization and vice-versa". A scientist unfamiliar to this approach and the particular challenge may interpret the proposed approach to be error-free.
- For the paired RNAseq/ATACseq approach, the authors must provide the data referenced in the supplementary materials and methods: "we acquired a transcription factor (TF) motif by cells matrix from the authors".
- The authors should provide rationale for why they chose model parameters and model architecture. A specific comment on why they selected a latent dimensionality of 50 would be helpful.
- The authors vary the "level of supervision" in figure 2. This is an extremely important point that should be both defined more clearly and highlighted more in the text. A big strength of the authors' approach is that "supervision" is not needed. Also, based on my understanding, "supervision" refers to having matched cells. Traditionally, "supervision" would more likely refer to a "supervised" learning algorithm, which this is not. Also, I understand how "supervision" is toggled for the autoencoder approach (anchor loss) but it is not clear how the "supervision" is varied for DCCA. Also, if DCCA supervision 0% is performed after 100 random samplings of the same label, shouldn't figure 2A have error bars?
- It appears that the authors include their processed gene expression matrix for the naïve CD4+ T cell analysis in their supplementary code. This is great! Please confirm this and also add access details in the supplementary methods.
- The co-association matrix (fig 3b) demonstrates consensus across clustering methods. Which methods? How many iterations? I do not see if this is ever defined.
- More methodological details are required for this statement: "Differential gene expression and GO enrichment analysis indicated that one cluster corresponded to quiescent cells while the other was poised for activation". Also, this is referenced as Fig 2c-d. This should probably actually reference Fig 3c-d.
- For figure 3G, bars or shading should be placed around each of the two populations to represent variance. Variance would provide insight into how stable and distinct the two groups are. What cluster analysis was used for the chromatin data?
- The supplement section "Autoencoder training on chromatin images for validation section" references a Figure 3H that does not exist.
- The dotted line in Figure 4C is not always a suitable "best guess" control. The authors should

randomly permute class labels and retrain the random forests to get a better null baseline.

- A simple statistical test comparing the two distributions in Figure 4H should be applied to determine significance.
- Gene names should be italicized.
- I don't know what the arrows and gene names represent in Figure 4I.
- In the methods: "Images were processed and further analyzed using custom programs in Fiji and R." – these custom programs should be provided
- The authors should specify that they use variational autoencoders instead of just saying autoencoders.
- How were the images in Figure S4 selected? Randomly? Please specify.

This is a signed review.

Gregory Way, Ph.D.
Postdoctoral Associate
Imaging Platform
Broad Institute of MIT and Harvard
415 Main Street
Cambridge, MA 02140

Reviewer #3 (Remarks to the Author):

In this paper, the authors propose a method for integration of heterogeneous datasets by deep learning. The problem that is addressed computationally in this paper is the joint analysis of data from different sources: if we acquire data using different techniques (e.g. transcriptomic, genomic or image data), they can in most cases not be obtained for the same cellular populations. The idea developed in this paper is to use autoencoders to map each of these data sources to a latent space and back, which means that we infer the rule according to which data in the latent space relates to data in the original space in both directions. The autoencoders can therefore be used to generate artificial data that has not been acquired. The authors validate their method on a dataset, where two data sources are available (scRNAseq and ATAC-seq) and show an application where they translate between scRNAseq and chromatin images.

The paper is well written. It is certainly rather unusual for the readers of Nature Communications, but the text is technically sound and the data is convincing. The problem addressed in the paper is of high relevance. This kind of approaches is highly controversial and will contribute to an ongoing discussion in the field, that usually leads to very emotional reactions: how much can we actually predict? Is it still science if we predict measurements instead of taking them? How valid are the conclusions that can be drawn from predicted data? To which extend a different training set would influence the final conclusions? ...

This paper makes a significant contribution to this important discussion and also introduces an interesting piece of methodological work. However, I feel that the authors did not include a proper discussion, in particular regarding the limitations of the method. I also would like to see some further validation.

1. My first concern is that the method assumes that P_Z is identical for the different modalities. Let us assume that cells fall into different categories (different clusters in the latent space), and each experiment with different modality is made of various proportions of cells in these categories. If I understand correctly, the constraint on P_Z means that you would need identical or at least similar proportions of cells in each category to make the method work. Is this realistic? It certainly is for the ATACseq / scRNAseq example, because it is the same cellular population, but in general? Can the authors give an estimate about the similarity that is required for their method to work? In any case, this limitation should be discussed.
2. Related to this, it is not entirely clear to me what the underlying principle of the data translation

actually is. On page 3, it becomes clear that the auto-encoder for domain i is trained in such a way as to recapitulate the distribution in the latent space induced by domain j . This is basically the coupling mechanism between domains (or modalities) without which it would be impossible to translate between domains. Let us now assume that for modality 1, we capture a number of marker genes (as done here), and in modality 2, we take an unrelated cell measurement (e.g. cell cycle). If the proportions are comparable, the network would link the expression of the unrelated marker genes to the readout in modality 2 and basically do a completely wrong association. What would the authors do to prevent such a behavior? I think the reader needs to understand what kind of mappings can be learned, what the underlying principle is and what the limitations of the proposed methodology are.

3. There is a number of papers that actually propose similar strategies, and some of them are mentioned in the introduction. I am surprised that there were not more benchmarking results on the scRNAseq / ATACseq data set. The authors should include more methods in this benchmark.

4. On page 4, the authors show first the clustering results they obtain from a number of known markers, leading to 2 groups as expected (naive and activated). Then they turn to imaging experiments and find also two groups from Figure 3.g. But while they validate their clustering approach for the scRNAseq data, they do not the same for the imaging data. And indeed, Figure 3.f seems not that convincing. There is clearly an overall shift between central vs. peripheral chromosome density, but the lower of the two clusters also contains a subcluster that could have been assigned to the upper one. An objective method to cluster the image data is required here, and the differences in clustering with respect to the final output might also be evaluated.

5. Related to point 4: not all cellular properties suggest grouping of cells in distinct states: migration speed, cell size and many other physical properties are continuous variables. This would be also an interesting point to discuss.

Altogether, I find this article interesting, and I vote for publication after revision.

Thomas Walter.

Minor comments:

1. The authors should also reference the publications [1], [2].
2. Figure 2 is not very readable, as the difference between solid lines and dashed lines does not appear very clearly in the legend.
3. I did not understand the impact of the level of supervision. What does it mean in practice that the "autoencoder model trained with just 25% supervision ... has similar performance to fully supervised DCCA" ? What do you get in practice from this improvement?
4. Page 4, Figure 2c-d is referenced, but it should be Figure 3c-d.

References

- [1] Mayer, I.; Josse, J.; Raimundo, F.; Vert, J.-P. MissDeepCausal: Causal Inference from Incomplete Data Using Deep Latent Variable Models. 2020.
- [2] Liu, J.; Huang, Y.; Singh, R.; Vert, J.-P.; Noble, W. S. Jointly Embedding Multiple Single-Cell Omics Measurements. In 19th international workshop on algorithms in bioinformatics (WABI 2019); Huber, K. T., Gusfield, D., Eds.; Leibniz international proceedings in informatics (LIPIcs); Schloss Dagstuhl–Leibniz-Zentrum fuer Informatik: Dagstuhl, Germany, 2019; Vol. 143, pp 10:1-10:13. <https://doi.org/10.4230/LIPIcs.WABI.2019.10>.

Reviewer #1: Comments and Responses

This paper describes an autoencoder for domain adaptation, which is also similar to the problem of batch correction. Both of these problems require an alignment of datasets, and has been characterized by many papers as manifold alignment. Moreover, autoencoders [REF7], and VAEs [REF5] have been used for this problem several times. Moreover, there is a whole set of works using cycleGANs [REF7, REF3] for both cross modal data alignment. In addition, there are approaches that use diffusion geometry and manifold learning methods for this purpose [REF7] showing results on single cell RNA-seq and ATAC-seq like this manuscript. However, the authors seem completely unaware of this body of literature, do not cite a single one of these papers, and don't compare the methods.

- [REF1] Liu R, Zou J, and Balsubramani A. Learning Transport Cost From Subset Correspondence. ICLR, 2020
- [REF2] Makhzani A, Shlens J, Jaitly N, Goodfellow I, Frey B. Adversarial Autoencoders. arXiv:151105644 [cs]. May 2016
- [REF3] Amodio M, Krishnaswamy S. MAGAN: Aligning Biological Manifolds. ICML 2018
- [REF4] Amodio M, Krishnaswamy S. Image-To-Image Translation by Transformation Vector Learning CVPR 2019
- [REF5] Lopez et al. Deep generative models for single-cell transcriptomics, Nature Methods 2018
- [REF6] Stanley et al. Harmonic Alignment, SIAM Data Mining 2020
- [REF7] Amodio et al. Exploring single-cell data with deep multitasking neural networks. Nature Methods 2019
- [REF8] Zhu et al. Unpaired Image-to-Image Translation using Cycle-Consistent Adversarial Networks ICCV 2017

For all of these methods the authors need to formulate comparisons, put their work in context and describe conceptual differences against. Moreover the authors need to relate their adversarial setup to an adversarial autoencoder [REF2]. One other point to consider is that the densities may be different in the different modalities. Alignment of geometries over densities is considered in [REF6]. The only point of impact may be the exact modes that they are aligning, spatial data with non-spatial sequencing data. Spatial sequencing measurements does require convolutional layers which also some of the above references use especially [REF8, REF4], but image feature extraction on single cell data may be different than natural image feature extraction. This is the point which needs to be explored in great detail which this manuscript does not do.

We thank the reviewer for these helpful comments and appreciate the related works pointed out by Reviewer 1. We carefully updated the main text to describe them as well as how they differ from our method (see below).

We would like to emphasize that a key technical contribution of our method is that we are performing cross-modal alignment *in the latent space* rather than in the original data space as in the referenced methods based on GANs. This has several advantages compared to existing approaches that perform cross-modal alignment:

- (A) Performing cross-modal alignment in the latent space is scalable to more than two modalities, as shown in Figure 1 in our paper. Additional modalities can be integrated by training a single autoencoder to align the new modality with the existing ones in the latent space. Existing methods for cross-modal alignment have limited scalability to multiple modalities, since a new model must be built to translate between every pair of modalities.
- (B) Beyond scalability, performing cross-modal alignment in the latent space enforces a notion of *global consistency* between the different modalities, which is missing from existing methods that consider cross-modal alignments between every pair of modalities. Consider an application of a CycleGAN-derived model [REF8] to translate between three modalities (A, B, C) by training three pair-wise models ($A \leftrightarrow B, B \leftrightarrow C, A \leftrightarrow C$). Note that the point obtained from directly mapping $A \rightarrow B$ may differ from the point obtained from mapping from $A \rightarrow C \rightarrow B$. In practice, this means that biological predictions are generally not consistent depending on how the cross-modal translations are performed. On the other hand, our method enforces this type of consistency by construction and can be used to build a biological model that aligns multiple modalities of data (e.g., images, RNA-seq, ATAC-seq, etc.) with consistent predictions across modes.
- (C) Since our method learns a joint latent space to perform cross-modal alignment, downstream data analysis such as clustering can be performed jointly across all of the modalities within the latent space. This is not possible with existing cross-modal methods that perform alignment directly in the input space.

In terms of comparison to batch correction methods [REF5, REF7], our approach is an extension to the cross-modal setting. The batch-correction methods enforce alignment in the latent space between different batches of data, which ensures that analysis of the latent representation of the data is disentangled from batch-dependent effects. Similarly, our method enforces alignment in the latent space between different modalities of data. However, the key conceptual difference is the nature of the transformation, which leads to key methodological differences:

- (D) In batch-correction, one assumes that the data structure of the different batches is similar, such that they can be mapped from the input space to the latent space using similar transformations. This is naturally enforced by using a common encoder/decoder model to transform and align all of the samples, with batch identity being input to the model as an additional variable. However, for cross-modal alignment between very different modalities (e.g., imaging data and RNA-seq data as considered in our paper) or modalities with different dimensionality (e.g., RNA-seq and ATAC-seq), the transformations cannot be assumed to be similar. Therefore, cross-modal alignment requires implementing and training a separate autoencoder for each data modality.
- (E) Since the transformation from the input spaces to the latent space is performed by a different autoencoder for each data modality, cross-modal alignment requires additional constraints on the alignment between different modalities. For example, our method proposes additional regularization techniques to constrain the way that the different modalities are mapped to and aligned in the latent space (see the section on “Incorporating prior knowledge” in our paper).

In the following, we provide a brief summary of each of the prior methods suggested by Reviewer 1 and underscore the conceptual differences of our method.

- **Zhu et al. Unpaired Image-to-Image Translation using Cycle-Consistent Adversarial Networks ICCV 2017.** CycleGAN is perhaps the most prominent of unsupervised cross-modal alignment methods. The method trains two translation networks on

discriminative (GAN) losses to align the translated data with the original data in each modality and also uses a cycle-consistency loss to ensure that translation between the two modalities is consistent, i.e., translating a point from domain $A \rightarrow B \rightarrow A$ yields the same original point in domain A . However, this cross-modal translation approach, which aligns distributions in two input spaces rather than one latent space, is limited compared to our method as described in (A), (B), and (C) above. Extensions of CycleGAN such as StarGAN (Choi et al, CVPR 2018) have been proposed for image-to-image translation between multiple modalities. StarGAN uses a common encoder/decoder model for all of the modalities, but this approach assumes that the input representations are similar and bears resemblance to the batch-correction / domain adaptation methods. Thus it differs from our approach as described in (D), (E) above.

- **Amodio M, Krishnaswamy S. MAGAN: Aligning Biological Manifolds. ICML 2018.** This paper presents an extension of CycleGAN with additional translation cost to encourage certain alignments between the data. Similar to CycleGAN, it is still limited compared to our method as described in (A), (B), (C) above.
- **Amodio M, Krishnaswamy S. Image-To-Image Translation by Transformation Vector Learning CVPR 2019.** This method for cross-domain alignment replaces the cycle-consistency loss of CycleGAN with “vector arithmetic consistency” in a common latent space as learned by a Siamese network framework. The alignment is performed in one of the input domains, which subjects it to the same limitations as described in (A), (B), (C) above. Moreover, the Siamese network framework is specific to image-to-image translation (e.g., translation between modalities with similar representations).
- **Liu R, Zou J, and Balsubramani A. Learning Transport Cost From Subset.** This method for cross-domain alignment is based on Sinkhorn optimal transport, where side information in the form of paired samples is used to learn the transport cost function. This method differs fundamentally from our method in that the solution, the transport plan, is a soft-matching between data points in the two modalities. Since it operates on pairs of modalities and does not align data in a latent space, this method would still be subject to the same limitations as described in (A), (B), (C) above.
- **Lopez et al. Deep generative models for single-cell transcriptomics, Nature Methods 2018, and Amodio et al. Exploring single-cell data with deep multi-tasking neural networks. Nature Methods 2019.** These works propose variants of variational autoencoders and autoencoders respectively for aligning data batches in the latent space. Lopez et al. aligns batches in the latent space using the variational lower bound objective of a conditional variational autoencoder, and Amodio et al. aligns batches in the latent space using maximum mean discrepancy (MMD) loss. Our work differs from these works as described in (D) and (E) above. Additionally, we use a discriminative approach to align modalities in the latent space rather than a variational loss or MMD loss.
- **Stanley et al. Harmonic Alignment, SIAM Data Mining 2020.** This work uses diffusion maps to learn the data manifolds and aligns them by considering correlations between the diffusion coordinates, which is fundamentally different from our deep learning-based method. This approach is primarily applicable to non-imaging data where the construction of a diffusion map based on a Gaussian kernel can accurately capture the data manifold.

- **Makhzani A, Shlens J, Jaitly N, Goodfellow I, Frey B. Adversarial Autoencoders.** [arXiv:151105644 \[cs\]](https://arxiv.org/abs/1511.05644). May 2016. Similar to our work, Makhzani *et al.* propose aligning the latent space of an autoencoder using adversarial training (discriminative loss) for generative modeling, but they do not leverage this technique for data integration and/or translation. Inspired by their work, our method extends their idea to perform adversarial training in the latent space for cross-modal data alignment.

With regards to updating the main text, we added [REF6] to our list of references in the introduction, since this method can be applied to different modalities with the same representation (i.e., different sequencing measurements) but does not extend to imaging data. In addition, in the introduction we also added the following sentences describing related work covering the suggested references [REF1-5, REF7-8]:

“Several works have proposed using autoencoders for domain adaptation (in particular batch correction) in the context of biological data [16, 17]. Different from these works, our method uses autoencoders to integrate and translate between different data modalities that may have very different representations. A separate line of work has proposed using neural networks to directly translate between pairwise modalities in an unsupervised manner [18, 19] or with side information [20, 21]. These methods tend to focus on modalities with similar representations (e.g., image-to-image-translation) and directly translate between pairs of modalities without learning a common latent representation of the data. In contrast, our work maps each data distribution to a common latent distribution using an autoencoder. This not only enables data integration and translation between arbitrary modalities in a globally consistent manner, but, importantly, it also enables performing downstream analysis such as clustering across multiple modalities at once. Other work has proposed coupled autoencoders to translate between paired biological data [22], which differs from our method that does not require paired data. Building on Makhzani *et al.* [23], we align the latent space of an autoencoder using adversarial training and leverage this technique for data integration and/or translation.”

Reviewer #2 Comments and Responses

Yang et al describe an exciting and novel approach to integrate single-cell imaging and gene expression data. The primary innovation is the autoencoder training procedure and the new loss functions to incorporate markers shared among datasets (discriminative loss) and paired sample groups (anchor loss). The authors first benchmark their model in a paired gene expression and ATACseq dataset. They observe that their approach outperforms deep canonical correlation analysis based on two evaluation metrics. Next, the authors apply their approach to a dataset with gene expression and DAPI image measurements. The authors demonstrate, through a series of anecdotes based on independent clustering, that DAPI images and gene expression data share common signal. Overall, the manuscript is very interesting and timely, but I have several major and minor concerns that if addressed, will improve the manuscript.

We thank the reviewer for these positive comments.

Major Concerns

- There are several reporting and methodology concerns that I elaborate on in the minor concerns below. Individually, each of these concerns are mostly minor, but taken together, the many incomplete methods and reporting causes a major concern.

We thank the reviewer for the constructive comments. Below, we provide a point-by-point response to each comment.

The authors need to expand their discussion. An emphasis on limitations of their approach should be a primary focus.

We have expanded our discussion with the following paragraph, which addresses several of the reviewer’s questions below and clarifies the limitation of the method / how we envision it being used: “While we used our method to align RNA-seq and imaging datasets, we have presented a general framework that can be adapted to numerous other biological problems. As indicated in Figure 1, our framework can be used to integrate datasets of different modalities simply by incorporating autoencoder architectures tailored to those modalities. For example, Hi-C data could be integrated using a graph neural network and multi-channel cell images using a convolutional neural network with different input channels. Also, while we focused on aligning datasets each containing two distinct clusters, our method can be applied to datasets with other distributions as long as the samples are taken from the same cell population. For example, in applications where there are no clear clusters in the datasets, our method can be used to align continuous markers between datasets by conditioning the adversarial loss on the values of the continuous marker (Equation 3). In applications where there is some shared signal between modalities as well as signal that is individual to each modality, our model can be extended by introducing a subset of latent dimensions that is specific to each modality. An important consideration, however, is that while our method can be applied for data integration and cross-modal alignment in generic contexts, depending on the data distributions, there may be multiple alignments that satisfy the same objective function. Additional constraints (in the form of prior knowledge) should be added to these models where possible to enforce alignments that are biologically accurate. Overall, we envision an iterative process of biological discovery where our predictive model is used for hypothesis generation, the hypotheses are validated (or disproved) experimentally, and the new experimental results now serve as additional data (prior knowledge) for improving the alignment of the model.”

Figure 1: Example of cross-modal alignment between RNA-seq and ChIP-seq data of mouse embryonic stem cells (Figure 7 from our earlier non-archival workshop paper [27])

In the main experiment, the authors used a nice dataset that appears to have clean separations between two groups in both imaging and expression data modalities. What happens to their approach in the much more common scenario where there are no clear groupings or even if there are different numbers of groups between different data modalities?

While we used our method to align RNA-seq and imaging datasets with two distinct clusters, we have presented a general framework that can be adapted to numerous other biological problems as long as the different datasets are sampled from the same underlying distribution. In applications where there are no clear clusters between the datasets, one could align continuous markers between datasets by conditioning the adversarial loss on continuous marker values; see what is now Equation 3 in the main text. We have clarified this point in the methods section, under the discriminative loss subsection by adding the following sentences: “This approach is valid for both discrete and continuous values of the cluster/marker y . For example, in [27], this approach was used to align a continuous differentiation marker between RNA-seq and ChIP-seq data.” To further illustrate this point, Figure 1 above (copied from our earlier non-archival workshop paper [27]) shows the example of aligning a continuous marker (in this case a differentiation marker) between RNA-seq and ChIP-seq data from mouse embryonic stem cells using our approach, where there are no clear clusters in either modality. This use case of our method is now also addressed in the discussion section of the revised manuscript as follows: “Also, while we focused on aligning datasets each containing two distinct clusters, our method can be applied to datasets with other distributions as long as the samples are taken from the same cell population. For example, in applications where there are no clear clusters between the datasets, our method can be used to align continuous markers between datasets by conditioning the adversarial loss on the values of the continuous marker (Equation 3).”

What about more complicated imaging assays beyond DAPI staining?

Our method can be applied to more complicated imaging assays by introducing an autoencoder architecture that is tailored to that particular data representation. For example, a multi-channel cell image can be captured using a convolutional neural network with the same number of channels in its first layer of convolutional filters. In the revised manuscript we clarified this point in the discussion section as described in the added paragraph above: “For example, Hi-C data could be integrated using a graph neural network and multi-channel cell

images using a convolutional neural network with different input channels.”

How do the authors propose to use their method to make biological discoveries?

We thank the reviewer for this helpful question. We envision an iterative process of biological discovery, where our predictive model is used for hypothesis generation, the hypotheses are validated (or disproved) experimentally, and the new experimental results now serve as additional data (prior knowledge) for improving the alignment of the model. We have added the following sentences to the discussion as described above to address this question: “An important consideration, however, is that while our method can be applied for data integration and cross-modal alignment in generic contexts, depending on the data distributions, there may be multiple alignments that satisfy the same objective function. Additional constraints (in the form of prior knowledge) should be added to these models where possible to enforce alignments that are biologically accurate. Overall, we envision an iterative process of biological discovery where our predictive model is used for hypothesis generation, the hypotheses are validated (or disproved) experimentally, and the new experimental results now serve as additional data (prior knowledge) for improving the alignment of the model.”

Don’t we expect certain signal to be unique to each modality? How can we use this approach to determine which signals are unique?

While we assumed in this particular instance that the distributions of the two datasets are the same, the method can also be applied to the case where there is a shared subset of latent dimensions between modalities and a subset of latent dimensions that is specific to each modality. We have clarified this in the section describing the model as follows: “Note that the assumption that each X_i is obtained via a deterministic function of Z implies that the latent distribution of each dataset is the same. However, by including the noise variables N_i as in Equation (1), our method extends to the case where only a subset of latent dimensions is shared between the different modalities and the remaining dimensions are specific to each modality.” To further emphasize this point, the following sentence has also been added to the discussion as mentioned above: “In applications where there is some shared signal between modalities as well as signal that is individual to each modality, our model can be extended by introducing a subset of latent dimensions that is specific to each modality. ”

- The authors include a training and testing set in the ATACseq + gene expression application. It is not clear if the authors use a holdout test set in the image-expression translation evaluation. The authors note “We held out 10% of the data as test data and trained until the reconstruction loss on the test data was smaller than 10^{-3} ”. This is not a true holdout set. Also, the authors do not note if their evaluations are based on the full dataset used for training.

In the revised manuscript, we have now clarified this as follows: “Consistent with other methods used for data integration and translation in the biological domain, where the goal is to provide a matching between samples in the observed datasets [12], our evaluation is based on the full dataset used for training rather than a held-out evaluation set.”

Minor Concerns

- Some sentences convey more confidence than the results warrant. For example, a sentence in the second paragraph of the introduction states: “In particular, our framework can be applied to integrate and translate imaging and sequencing data, which cannot yet be obtained

experimentally in the same cell, thereby providing a methodology to predict the genome-wide expression profile of a particular cell given its chromatin organization and vice-versa". A scientist unfamiliar to this approach and the particular challenge may interpret the proposed approach to be error-free.

In the revised manuscript, we reworded the sentence to emphasize hypothesis generation: "In particular, our framework can be applied to integrate and translate imaging and sequencing data, which cannot yet be obtained experimentally in the same cell, thereby providing a methodology for hypothesis generation to predict the genome-wide expression profile of a particular cell given its chromatin organization and vice-versa."

- For the paired RNAseq/ATACseq approach, the authors must provide the data referenced in the supplementary materials and methods: "we acquired a transcription factor (TF) motif by cells matrix from the authors".

We now provide this data as supplementary Data S1: "Data S1: Transcription factor motif by cells matrix for ATAC-seq data from A549 cells."

- The authors should provide rationale for why they chose model parameters and model architecture. A specific comment on why they selected a latent dimensionality of 50 would be helpful.

We thank the reviewer for this helpful comment. In the revised manuscript, we added the following sentences to address this point: "In practice, the model architecture of each autoencoder is selected based on the input data representation (e.g., fully-connected network for gene expression data and convolutional network for images). The dimensionality of the latent distribution is a hyperparameter that is tuned to ensure that the autoencoders are able to reconstruct the respective data modalities well. For sequencing data, PCA can be used to obtain an initial estimate of the intrinsic dimensionality of the data, which can then be fine-tuned by analyzing the reconstruction loss of the model. For imaging data the reconstruction quality can also be assessed qualitatively (see Figure S5) and a variational autoencoder with a small weight on the KL-divergence regularization term can be used to improve image generation quality."

- The authors vary the "level of supervision" in figure 2. This is an extremely important point that should be both defined more clearly and highlighted more in the text. A big strength of the authors' approach is that "supervision" is not needed. Also, based on my understanding, "supervision" refers to having matched cells. Traditionally, "supervision" would more likely refer to a "supervised" learning algorithm, which this is not. Also, I understand how "supervision" is toggled for the autoencoder approach (anchor loss) but it is not clear how the "supervision" is varied for DCCA. Also, if DCCA supervision 0% is performed after 100 random samplings of the same label, shouldn't figure 2A have error bars?

We thank the reviewer for this comment. Indeed, we used the term "supervision" to refer to the availability of matched data points, a type of prior knowledge. As per the reviewer's comment on the traditional use of the word "supervision", we replaced this term by "paired samples/cells/data". In the section on "Incorporating prior knowledge", we define two types of prior knowledge: (a) using shared markers/clusters, and (b) anchor loss (knowing which data points are paired with each other). On paired RNA-seq and ATAC-seq samples we used the anchor loss to ensure that matching data points are close to each other in the latent space. To

better explain these points, we added the following sentences in the main text: “While paired data was only used to evaluate the accuracy in Fig. 2a-b, Fig. 2c-e explore the setting in which paired data on a fraction of samples is used for training. Although paired data is not necessary for our method, such prior knowledge can be incorporated using the anchor loss described above, which ensures that paired samples are close in the latent space. Fig. 2c-d show that our autoencoder model outperforms DCCA when trained on varying amounts of paired data. In fact, as shown in Fig. 2e, our autoencoder model trained with just 25% of the paired samples has similar performance to DCCA trained on all (i.e. 100%) of the paired samples, thereby indicating that our method is practical and competitive also in the setting where some paired data is available.” In addition, in SI Appendix, Materials and Methods, we write: “For both our cross-modal autoencoder method and DCCA, we explored the use of samples whose pairing is known between the two domains (i.e., anchored cells in both datasets), which is available in some applications. To make use of the pairing information in our cross-modal autoencoder model, we included an additional term in the loss function corresponding to the mean absolute error between the paired training points in the latent space.”

We clarified the question regarding what DCCA with 0% supervision means in SI Appendix, Materials and Methods, as follows: “While our method based on autoencoders does not require paired samples, DCCA does. In order to train DCCA with 0% paired samples, we randomly generated paired samples using the treatment time labels of the cells as follows. For each point with a particular treatment time label, we sampled 100 random points with the same label to use as its paired samples.” Since all the artificial pairings were used for training, we obtained a single performance number and thus do not have error bars in Figure 2a.

- It appears that the authors include their processed gene expression matrix for the naïve CD4+ T cell analysis in their supplementary code. This is great! Please confirm this and also add access details in the supplementary methods.

In the original submission, we had included the differential expression analysis results in SI Appendix. In the revised submission, we now include also the processed gene expression matrix as part of Data S2. We updated the description of Data S2 (previously Data S1) accordingly:

”Data S2: Cluster label assignments based on single-cell RNA-seq for PBMC cells. Cluster label assignment for naïve CD4+ T-cells based on single-cell RNA-seq. Differential expression of genes between quiescent and poised naïve CD4+ T-cells. Gene expression matrix corresponding to naïve CD4+ T-cells.”

- The co-association matrix (fig 3b) demonstrates consensus across clustering methods. Which methods? How many iterations? I do not see if this is ever defined.

The procedure for obtaining the co-association matrix was described in the ”Gene expression analysis of naïve CD4+ T-cells” section in the Methods section; namely:

”To test the robustness with respect to different clustering methodologies, we also used k-means, Gaussian mixture models and spectral clustering based on a k-nearest neighbor graph with $k \in \{10, 20, 50, 100\}$ to cluster the data. We performed 100 different initializations for each method and computed the co-association matrix, which quantifies how often each pair of cells was clustered together; the result is shown in Fig. 3b.”

In order to make these methodological details easier to find, in the revised manuscript we

added them also to the caption of Fig. 3b as follows: "Gene expression data was clustered using k -means, Gaussian mixture models and spectral clustering based on a k -nearest neighbor graph with $k \in \{10, 20, 50, 100\}$ with 100 initializations for each method."

- More methodological details are required for this statement: "Differential gene expression and GO enrichment analysis indicated that one cluster corresponded to quiescent cells while the other was poised for activation". Also, this is referenced as Fig 2c-d. This should probably actually reference Fig 3c-d.

We thank the reviewer for this comment; we have now included a more detailed discussion of this in the paper as follows: "Specifically, we observed that one of the two clusters of naive CD4+ T-cells contained "immune response" and "cell activation" as one of the top significant GO terms as well as a well-known activation marker IL32 as one of the differentially expressed genes."

We also fixed the references to the figures; thanks for noticing this.

- For figure 3G, bars or shading should be placed around each of the two populations to represent variance. Variance would provide insight into how stable and distinct the two groups are.

Good suggestion; we now provide a plot that includes standard deviation shading (Figure 3g in the revised manuscript and Figure 2 below). We updated the figure caption of Figure 3g accordingly by adding "(standard deviation represented by shading)".

Figure 2: Updated plot of average chromatin signal in concentric spheres with increasing radii for central (green) and peripheral (blue) clusters with shading showing the standard deviation.

What cluster analysis was used for the chromatin data?

We thank the reviewer for this question. The chromatin data was clustered using hierarchical clustering with complete linkage based on the distance matrix obtained from 1- Spearman's correlation. This has been clarified in the caption of Figure 3g as follows: "The features were clustered using hierarchical clustering with complete linkage based on the distance matrix obtained from 1-Spearman's correlation. "

- The supplement section "Autoencoder training on chromatin images for validation section" references a Figure 3H that does not exist.

The reference should have been to Fig. 4h. Thanks for pointing this out; we fixed this accordingly.

- The dotted line in Figure 4C is not always a suitable “best guess” control. The authors should randomly permute class labels and retrain the random forests to get a better null baseline.

We did not mean for the dotted line to be a competitive baseline. We meant for it to represent a random guess based on evenly-distributed classes. This has been clarified in the figure caption by adding: “The dotted line represents random guessing based on evenly-distributed classes.”

- A simple statistical test comparing the two distributions in Figure 4H should be applied to determine significance.

Using two-sided Welch’s t-test for difference in means gives a p-value $< 2.2 \times 10^{-16}$. We have added this p-value to the caption in Figure 4h.

- Gene names should be italicized.

Thank you for pointing this out; we fixed this accordingly.

- I don’t know what the arrows and gene names represent in Figure 4I (upregulated / down-regulated).

The arrows represent which genes are upregulated and which are downregulated as predicted by our model. We have clarified this in the caption of Figure 4i: “The up and down arrows represent which genes are upregulated and downregulated respectively as predicted by our model.”

- In the methods: “Images were processed and further analyzed using custom programs in Fiji and R.” – these custom programs should be provided

We have added the location of these programs in the section Data and code availability; namely: “The primary images and code are available at <https://github.com/uhrerlab/cross-modal-autoencoders>.”

- The authors should specify that they use variational autoencoders instead of just saying autoencoders.

We would like to clarify that our method is not strictly based on variational autoencoders. For example, the RNA-seq and ATAC-seq experiment uses regular autoencoders. As per the model section, the main objectives of the autoencoder training are (1) reconstruction loss and (2) discriminative loss between latent distribution and target latent distribution. Thus our choice of autoencoder regularization is more similar to that of adversarial autoencoders (Makhzani et al., 2015) than variational autoencoders. For the RNA-seq and image experiment, we use variational autoencoders with a very small weight on the KL divergence regularization term (i.e., 10^{-8}) to improve image generation quality. This has now been clarified in the text describing our model as follows: “For imaging data the reconstruction quality can also be assessed qualitatively (see Figure S5) and a variational autoencoder with a small weight on the KL-divergence

regularization term can be used to improve image generation quality.”

- How were the images in Figure S4 selected? Randomly? Please specify.

The images were selected randomly. In the revised manuscript, we updated the caption of Figure S4 (now Figure S5) to clarify this point by adding: “Images were selected randomly.”

This is a signed review. Gregory Way, Ph.D. Postdoctoral Associate Imaging Platform Broad Institute of MIT and Harvard 415 Main Street Cambridge, MA 02140

Reviewer #3 Comments and Responses

In this paper, the authors propose a method for integration of heterogeneous datasets by deep learning. The problem that is addressed computationally in this paper is the joint analysis of data from different sources: if we acquire data using different techniques (e.g. transcriptomic, genomic or image data), they can in most cases not be obtained for the same cellular populations. The idea developed in this paper is to use autoencoders to map each of these data sources to a latent space and back, which means that we infer the rule according to which data in the latent space relates to data in the original space in both directions. The autoencoders can therefore be used to generate artificial data that has not been acquired. The authors validate their method on a dataset, where two data sources are available (scRNAseq and ATAC-seq) and show an application where they translate between scRNAseq and chromatin images.

The paper is well written. It is certainly rather unusual for the readers of Nature Communications, but the text is technically sound and the data is convincing. The problem addressed in the paper is of high relevance. This kind of approaches is highly controversial and will contribute to an ongoing discussion in the field, that usually leads to very emotional reactions: how much can we actually predict? Is it still science if we predict measurements instead of taking them? How valid are the conclusions that can be drawn from predicted data? To which extend a different training set would influence the final conclusions? ... This paper makes a significant contribution to this important discussion and also introduces an interesting piece of methodological work. However, I feel that the authors did not include a proper discussion, in particular regarding the limitations of the method. I also would like to see some further validation.

We thank the reviewer for the positive and constructive comments.

1. My first concern is that the method assumes that P_Z is identical for the different modalities. Let us assume that cells fall into different categories (different clusters in the latent space), and each experiment with different modality is made of various proportions of cells in these categories. If I understand correctly, the constraint on P_Z means that you would need identical or at least similar proportions of cells in each category to make the method work. Is this realistic? It certainly is for the ATACseq / scRNAseq example, because it is the same cellular population, but in general? Can the authors give an estimate about the similarity that is required for their method to work? In any case, this limitation should be discussed.

Our method assumes that the datasets are collected from the same (or similar) cell populations, such that the underlying latent distributions of the datasets are the same. This is a realistic assumption for many applications – for example, it is common for cell samples to be collected from the same tissue and then separated for different downstream experiments and analysis. The method could also be adapted to the case where there are some signals that are shared between data modalities and other signals that are distinct. This could for example be implemented using a subset of latent dimensions that are shared between modalities and the remaining latent dimensions be specific to each modality. In the revised manuscript, we clarified this point as follows in the section describing our model: “Note that the assumption that each X_i is obtained via a deterministic function of Z implies that the latent distribution of each dataset is the same. However, by including the noise variables N_i as in Equation (1), our method extends to the case where only a subset of latent dimensions is shared between the different modalities and the remaining dimensions are specific to each modality.”

2. Related to this, it is not entirely clear to me what the underlying principle of the data translation actually is. On page 3, it becomes clear that the auto-encoder for domain i is trained in such a way as to recapitulate the distribution in the latent space induced by domain j . This is basically the coupling mechanism between domains (or modalities) without which it would be impossible to translate between domains. Let us now assume that for modality 1, we capture a number of marker genes (as done here), and in modality 2, we take an unrelated cell measurement (e.g. cell cycle). If the proportions are comparable, the network would link the expression of the unrelated marker genes to the readout in modality 2 and basically do a completely wrong association. What would the authors do to prevent such a behavior? I think the reader needs to understand what kind of mappings can be learned, what the underlying principle is and what the limitations of the proposed methodology are.

We thank the reviewer for this comment. The main assumption required for our method is that the different datasets present separate “views” of the same underlying latent variables. We agree with the reviewer that arbitrary and incorrect mappings can be learned, which is why it is important to include prior knowledge if possible, e.g., using a discriminative loss to align common markers/clusters. We have added the following paragraph to the discussion section clarifying the limitations of our approach as well as providing possible extensions to our approach: “While we used our method to align RNA-seq and imaging datasets, we have presented a general framework that can be adapted to numerous other biological problems. As indicated in Figure 1, our framework can be used to integrate datasets of different modalities simply by incorporating autoencoder architectures tailored to those modalities. For example, Hi-C data could be integrated using a graph neural network and multi-channel cell images using a convolutional neural network with different input channels. Also, while we focused on aligning datasets each containing two distinct clusters, our method can be applied to datasets with other distributions as long as the samples are taken from the same cell population. For example, in applications where there are no clear clusters in the datasets, our method can be used to align continuous markers between datasets by conditioning the adversarial loss on the values of the continuous marker (Equation 3). In applications where there is some shared signal between modalities as well as signal that is individual to each modality, our model can be extended by introducing a subset of latent dimensions that is specific to each modality. An important consideration, however, is that while our method can be applied for data integration and cross-modal alignment in generic contexts, depending on the data distributions, there may be multiple alignments that satisfy the same objective function. Additional constraints (in the form of prior knowledge) should be added to these models where possible to enforce alignments that are biologically accurate. Overall, we envision an iterative process of biological discovery where our predictive model is used for hypothesis generation, the hypotheses are validated (or disproved) experimentally, and the new experimental results now serve as additional data (prior knowledge) for improving the alignment of the model.”

While we used a discriminative loss to align common clusters in RNA-seq and imaging datasets, our framework can be adapted also when no clear clusters between the datasets are available. As pointed out in the revised discussion section above, this can be achieved by aligning continuous markers between datasets by conditioning the adversarial loss on the values of the continuous marker. To further illustrate this point, Figure 1 above (copied from our earlier non-archival workshop paper [27]) shows the example of aligning a continuous marker (in this case a differentiation marker) between RNA-seq and ChIP-seq data from mouse embryonic stem cells using our approach, where there are no clear clusters in either modality.

3. There is a number of papers that actually propose similar strategies, and some of them are mentioned in the introduction. I am surprised that there were not more benchmarking results on the scRNAseq / ATACseq data set. The authors should include more methods in this benchmark.

We thank the reviewer for this comment. In the revised manuscript, we now include additional benchmarking results on the scRNAseq / ATACseq data set. Namely, we compare our method to the (or one of the) most widely used methods for data integration “Seurat”. Briefly, this method assumes that the features across different modalities are the same and learns a shared embedding using canonical correlation analysis (CCA). The results of this additional benchmarking experiment are shown below in Figure 3 (in Figure 2 in the revised manuscript) along with the following brief description: ”In Fig. 2, we compare our cross-modal autoencoder framework to deep canonical correlation analysis (DCCA) [29], which determines a nonlinear transformation of the two datasets to maximize the correlation of the resulting representations, as well as to Seurat, a prominent method for biological data intergration of similar modalities [9,12]. Our autoencoder framework outperforms Seurat and is competitive with DCCA for integrating single-cell RNA-seq and single-cell ATAC-seq data both in terms of fraction of cells assigned to the correct cluster (Fig. 2a) as well as k -nearest neighbor accuracy (Fig. 2b).”

We also added the following accompanying text in SI Materials and Methods to describe this comparison and how we implemented Seurat: “We additionally compared our method against a popular method for data integration, Seurat [3,4]. Briefly, this method assumes that the features across different modalities are the same and learns a shared embedding using CCA based on this assumption. In order to apply Seurat to this particular dataset, we used the Seurat pipeline as follows: In order to obtain from ATAC-seq data a matrix that has the same features as the gene expression matrix, the ATAC-seq data was transformed into a gene activity matrix using the CreateGeneActivityMatrix function. We normalized and scaled the data using the NormalizeData and ScaleData functions. Finally, a shared CCA embedding was learned using the FindTransferAnchors functionality. Similar to our cross-modal autoencoder and DCCA, we

Figure 3: Comparison of our cross-modal autoencoder model, DCCA, and Seurat, a popular method for data integration.

used the inferred CCA embedding to quantify the method’s performance. Note that Seurat was fit using both training and test data, thereby giving Seurat an advantage over the other methods.”

4. On page 4, the authors show first the clustering results they obtain from a number of known markers, leading to 2 groups as expected (naive and activated). Then they turn to imaging experiments and find also two groups from Figure 3.g. But while they validate their clustering approach for the scRNAseq data, they do not the same for the imaging data. And indeed, Figure 3.f seems not that convincing. There is clearly an overall shift between central vs. peripheral chromosome density, but the lower of the two clusters also contains a subcluster that could have been assigned to the upper one. An objective method to cluster the image data is required here, and the differences in clustering with respect to the final output might also be evaluated.

We thank the reviewer for this helpful comment. To address this, we provided a careful

Figure 4: New supplementary figure for clustering of T-cell imaging data. This figure indicates that two clusters is an appropriate choice for the analysis using (a) average silhouette width, (b) gap statistic, (c) total within-cluster sum of square. (d) Alternative clustering using 1 - Pearson’s correlation matrix with average linkage. Green and blue colors represent original cluster labels.

analysis of number of clusters and robustness with respect to clustering methods. The results are provided above in Figure 4. We also added this figure in the revised supplement as Fig. S4. Using a variety of ways to evaluate the optimal number of clusters (average silhouette width, gap statistic and total within-cluster sum of square) all suggest that our dataset contains 2 clusters (see figures (a)-(c)). For clustering the imaging data we used hierarchical clustering with complete linkage based on 1-Spearman’s correlation to obtain the distance matrix. We chose this distance metric in order to avoid potential batch effects between samples. In order to test whether the results are robust to the choice of this metric, we redid the analysis using 1 - Pearson’s correlation and we used average linkage instead of complete linkage. As shown in figure (d), this alternative clustering method gives similar results.

5. Related to point 4: not all cellular properties suggest grouping of cells in distinct states: migration speed, cell size and many other physical properties are continuous variables. This would be also an interesting point to discuss. (continuous markers)

While we used our method to align RNA-seq and imaging datasets with two distinct clusters, we have presented a general framework that can be adapted to numerous other biological problems as long as the different datasets are sampled from the same underlying distribution. In applications where there are no clear clusters between the datasets, one can align continuous markers between datasets by conditioning the adversarial loss on continuous marker values; see what is now Equation (3) in the main text. We have clarified this point in the methods section, under the discriminative loss subsection by adding the following sentences: “This approach is valid for both discrete and continuous values of the cluster/marker y . For example, in [27], this approach was used to align a continuous differentiation marker between RNA-seq and ChIP-seq data.” To further illustrate this point, Figure 1 (copied from our earlier non-archival workshop paper [27]) shows an example, where a continuous marker (in this case a differentiation marker) was used to align RNA-seq and ChIP-seq data from mouse embryonic stem cells using our approach, where there are no clear clusters in either modality. We would be happy to include this example as additional validation as a supplementary figure.

Altogether, I find this article interesting, and I vote for publication after revision.
Thomas Walter.

Minor comments: 1. The authors should also reference the publications [1], [2].

Thanks for the suggestions. We have added [2] to our introduction and references. We feel that [1] is beyond the scope of our work; however, we would be happy to add it to our references if the reviewer could clarify its relation to our work.

2. Figure 2 is not very readable, as the difference between solid lines and dashed lines does not appear very clearly in the legend.

In the revised manuscript, we replaced Figure 2 in the main text with a new version to improve readability (by adding different marker types in addition to dashed lines). We included it below in Figure 5 for your reference.

3. I did not understand the impact of the level of supervision. What does it mean in practice that the “autoencoder model trained with just 25% supervision . . . has similar performance to fully supervised DCCA” ? What do you get in practice from this improvement?

Figure 5: Figure 2 with improved readability.

While in this particular application of integrating RNA-seq and ATAC-seq data, paired data for all samples is available, in other settings only some paired data might be available and much more unpaired data might be available (e.g. only some cells have both RNA-seq and ATAC-seq measured in the same cell). We wanted to understand the performance of our model in this setting. For example, our autoencoder model trained with 25% supervision considers the case where 25% of the cells in the dataset have paired information (e.g. ATAC-seq and RNA-seq collected in the same cell), while the remaining 75% of cells do not have such paired information. In Figure 2e we analyzed how much paired data our autoencoder framework would require in order to match the performance of DCCA trained using 100% paired data, showing that just 25% suffices. Thus our method is practical and competitive in settings where only partial paired data is available. We clarified this point in the main text as follows:

“While paired data was only used to evaluate the accuracy in Fig. 2a-b, Fig. 2c-e explore the setting in which paired data on a fraction of samples is used for training. Although paired data is not necessary for our method, such prior knowledge can be incorporated using the anchor loss described above, which ensures that paired samples are close in the latent space. [...] In fact, as shown in Fig. 2e, our autoencoder model trained with just 25% of the paired samples has similar performance to DCCA trained on all (i.e. 100%) of the paired samples, thereby indicating that our method is practical and competitive also in the setting where some paired data is available.”

4. Page 4, Figure 2c-d is referenced, but it should be Figure 3c-d.

Thank you, we fixed this accordingly.

References

- Mayer, I.; Josse, J.; Raimundo, F.; Vert, J.-P. MissDeepCausal: Causal Inference from Incomplete Data Using Deep Latent Variable Models. 2020.
- Liu, J.; Huang, Y.; Singh, R.; Vert, J.-P.; Noble, W. S. Jointly Embedding Multiple Single-Cell Omics Measurements. In 19th international workshop on algorithms in bioinformatics (WABI 2019); Huber, K. T., Gusfield, D., Eds.; Leibniz international proceedings in informatics (LIPIcs); Schloss Dagstuhl–Leibniz-Zentrum fuer Informatik: Dagstuhl, Germany, 2019; Vol. 143, pp 10:1-10:13. <https://doi.org/10.4230/LIPIcs.WABI.2019.10>.

REVIEWER COMMENTS

Reviewer #1 (Remarks to the Author):

Thank you for clearing up and improving the placement of your work within the broader literature. However, while the differences between your method and previous domain alignment techniques is conceptually established, the authors have not added any comparisons with other similar domain alignment techniques. Comparison to methods that do not work in the latent space would be useful in showing in what data the benefits of alignment in the latent space outweighs its limitations.

Since the two main validation experiments are done between two domains, comparisons to methods that operate in the data space are necessary to answer the question of whether alignment in the latent space is necessary or useful for fewer than 3 modalities.

The authors also did not address how the feature extraction priors affects the integration. For example, does a 3-layer convolutional encoder embed the same as a 3-layer dense encoder? Does using a 2-layer convolutional encoder change this? Alignment in the latent space has some conceptual benefits, but robustness of the integration to the encoder architecture seems like a crucial component in this framework, but this cannot be assessed without additional experiments and detail.

Reviewer #3 also brought up lack of benchmarking against similar methods. The addition of a Seurat comparison is certainly useful, it further emphasizes the lack of comparison to more comparable neural network based methods of alignment.

Reviewer #2 (Remarks to the Author):

The authors have adequately addressed most of the concerns I raised in the first round, and have improved the benchmarks, readability, and placement of the paper in the larger body of existing, similar literature. This paper presents an important contribution to the field. Thank you!

I do have a few additional comments to address. In order of importance:

1. GitHub Management: The addition of the GitHub resource is great to see. I also noticed that in the cross-modal-autoencoders repository, there is a link to preprocessing scripts at https://github.com/SaradhaVenkatachalapathy/Radial_chromatin_packing_immune_cells. This secondary repository should also be referenced in the manuscript. Additionally, both resources should be given an appropriate, open source license so that others can be instructed on how best to attribute and build off of the software this work has produced. Lastly, I strongly urge that these GitHub repositories be archived using an independent secondary service (e.g. Zenodo). Archival of the software provides an even stronger link to reproducibility in perpetuity.
2. Adding the usage note: "an iterative process of biological discovery where our predictive model is used for hypothesis generation" is quite vague. Presumably there could be hundreds of thousands of hypotheses that could be tested. What classes of hypotheses do the authors propose? Is it possible to provide a specific example about what the authors envision?
3. The authors discuss, but do not test, the addition of single-modality-specific latent features (Ni) to overcome the concern regarding single-modality-specific information. Is this indeed the case? If so, the authors should note that this feature of the architecture was not thoroughly vetted.
4. The method details for the differential gene expression / Gene Ontology enrichment analyses are still lacking. Specifically, what GO software was used? What was the gene background used to determine GO enrichment? Was all of GO included or only a subset? What method was used and what was the cutoff in determining differential expression?

This is a signed review. Gregory Way, Ph.D. Postdoctoral Associate Imaging Platform Broad Institute of MIT and Harvard 415 Main Street Cambridge, MA 02140

Reviewer #3 (Remarks to the Author):

The authors have addressed all the issues that I raised in my first review. In particular, they discuss conditions and limitations of the presented method, which will be very valuable for the scientific community. I therefore believe that this paper is going to make a fine contribution to the field.

Reviewer #1: Comments and Responses

Thank you for clearing up and improving the placement of your work within the broader literature. However, while the differences between your method and previous domain alignment techniques is conceptually established, the authors have not added any comparisons with other similar domain alignment techniques. Comparison to methods that do not work in the latent space would be useful in showing in what data the benefits of alignment in the latent space outweighs its limitations. Since the two main validation experiments are done between two domains, comparisons to methods that operate in the data space are necessary to answer the question of whether alignment in the latent space is necessary or useful for fewer than 3 modalities.

We thank the reviewer for the detailed comments on our work.

We further extended our computational performance comparison of prior work to include work that does not rely on a latent space. In particular, we now quantify the performance of CycleGAN, a prominent method for domain translation, which ensures that source samples are recovered back after mapping source samples to target domain and back to the source domain, and MAGAN, which is an extension of CycleGAN that additionally makes use of correspondences between domains. Both of these methods do not align the modalities in the latent space. We note that an advantage of learning a common latent space between modalities is that it enables clustering of cells in the common latent space as opposed to based on each modality (each modality may give a different clustering), which is often of interest to biologists who want to characterize cells into cell types based on the measured modalities. Fig. 1 below (respectively Fig. 2b,d in the revised main text) shows that our method outperforms CycleGAN and MAGAN. In particular, Fig. 1a compares models that do not use any correspondences (samples known to be paired) between the modalities (i.e., 0% supervision). Since MAGAN requires some correspondences for training (otherwise it corresponds to CycleGAN), it is not included in Fig. 1a. Fig. 1b compares our cross-modal autoencoder to other models (DCCA, CycleGAN, MAGAN) with varying amount of supervision (samples that are known to be paired) from 0% to 100% supervision. Note that MAGAN with 0% supervision corresponds to the CycleGAN model.

We also remark that we trained the same architecture that we used for our Image+RNA-seq model with the CycleGAN losses. However, this model did not produce realistic and diverse cell images after 2000 epochs, for various choices of hyper-parameters. This indicates that, in addition to the advantages of using a latent space described above, our cross-modal alignment approach based on aligning modalities using a single discriminator in a lower-dimensional latent space also provides additional stability and regularization for small biological imaging datasets compared to cross-modal alignment methods that operate using two discriminators in higher-dimensional input spaces.

We added the following accompanying text in SI Materials and Methods to describe the comparison with CycleGAN, MAGAN and our implementation on the paired RNA-seq and ATAC-seq data: "Finally, we compared our method against CycleGAN [5], a prominent deep learning method for domain translation, which ensures that source samples are recovered back after mapping source samples to target domain and back to the source domain. We used the code provided by the authors of CycleGAN at <http://github.com/junyanz/pytorch-CycleGAN-and-pix2pix> to translate ATAC-seq to RNA-seq and RNA-seq to ATAC-seq. We modified the architecture of

Figure 1: Comparison of our cross-modal autoencoder model, DCCA, Seurat, CycleGAN and MAGAN. (a) k -nearest neighbor accuracy for models trained on data with 0% supervision (paired samples). (b) k -nearest neighbor accuracy for models trained on data with 0-100% supervision.

the generator and discriminator networks to handle non-image data and match the architecture of our cross-modal autoencoder. In particular, the generator for translating ATAC-seq to RNA-seq consisted of a sequence of fully-connected layers with the following sizes: 815, 815, 815, 100, 50, 100, 2613, 2613, 2613. Similarly, the generator for translating RNA-seq to ATAC-seq consisted of a sequence of fully-connected layers with the following sizes: 2613, 2613, 2613, 815, 815, 100, 50, 100, 815, 815, 815. The discriminator model for ATAC-seq data took as input 815 features, followed by 815 hidden nodes and then 100 hidden nodes with a final output layer of size 1. The discriminator model for RNA-seq data took as input 2613 features, followed by 2613 hidden nodes and then 100 hidden nodes with a final output layer of size 1. All models used LeakyReLU as activation. The CycleGAN was trained for 2000 epochs with a learning rate of 0.0002 and batch size of 32. We evaluated the model only in terms of the k -nearest neighbor accuracy since the fraction of cells in the correct cluster was meant to evaluate the quality of the latent space. The k -nearest neighbor accuracy of the CycleGAN was computed in the original instead of the latent space since the model does not rely on the latent space for domain translation. Similarly, we compared our method against MAGAN [6], which has an additional correspondence loss term that ensures the measurements coming from the same sample should be close to each other. We trained MAGAN by providing 5%, 50% and 100% of paired samples in the training data for the correspondence loss.”

Accordingly, we have also updated the Figure caption of Fig. 2 in the manuscript and the section in the revised manuscript on model validation on paired single-cell RNA-seq and ATAC-seq data to reflect the addition of the two new methods (all changes are shown in red in the revised manuscript).

The authors also did not address how the feature extraction priors affects the integration. For example, does a 3-layer convolutional encoder embed the same as a 3-layer dense encoder? Does using a 2-layer convolutional encoder change this?

To demonstrate the robustness of our proposed data integration strategy, we trained models with varying numbers of latent dimensions, varying numbers of layers in the variational autoencoder (VAE), and different architecture choices of the image VAE (convolutional versus

fully-connected). We redid the analysis of Figure 3(b-d) for each of these models and added these new figures in the supplementary material (see Figures 2-3 below, corresponding to Figures S8 and S9 in the revised manuscript) to show that our data integration strategy is effective across different model architectures.

To reflect this analysis we added the following sentence to the revised main text: "In SI Appendix, Fig. S8-9, we show that our findings are robust to different architecture choices (fully-connected versus convolutional layers, number of layers, as well as latent space dimension)."

Alignment in the latent space has some conceptual benefits, but robustness of the integration to the encoder architecture seems like a crucial component in this framework, but this cannot be assessed without additional experiments and detail. Reviewer #3 also brought up lack of benchmarking against similar methods. The addition of a Seurat comparison is certainly useful, it further emphasizes the lack of comparison to more comparable neural network based methods of alignment.

We thank the reviewer for the comments. Our added experiments described above show that our proposed data integration strategy outperforms other comparable neural network based strategies that do not make use of a latent space and that our cross-model autoencoder is robust to different architecture choices (fully-connected versus convolutional layers, number of layers, as well as latent space dimension). In fact, our experiments indicate that the addition of a lower-dimensional latent space helped provide stability and regularization compared to cross-modal alignment strategies that operate directly in the high-dimensional input spaces and require discriminators in these high-dimensional spaces as compared to our discriminator which operates in the lower-dimensional latent space.

Reviewer #2 Comments and Responses

The authors have adequately addressed most of the concerns I raised in the first round, and have improved the benchmarks, readability, and placement of the paper in the larger body of existing, similar literature. This paper presents an important contribution to the field. Thank you!

I do have a few additional comments to address. In order of importance:

1. GitHub Management: The addition of the GitHub resource is great to see. I also noticed that in the cross-modal-autoencoders repository, there is a link to preprocessing scripts at https://github.com/SaradhaVenkatachalapathy/Radial_chromatin_packing_immune_cells. This secondary repository should also be referenced in the manuscript. Additionally, both resources should be given an appropriate, open source license so that others can be instructed on how best to attribute and build off of the software this work has produced. Lastly, I strongly urge that these GitHub repositories be archived using an independent secondary service (e.g. Zenodo). Archival of the software provides an even stronger link to reproducibility in perpetuity.

We have added the secondary repository in the Data and Code Availability section. Regarding licenses, we have added the MIT Open Source License to the main repository. In terms of archiving using an independent secondary service, we will also deposit the data with an appropriate license on Zenodo upon publication of the manuscript.

2. Adding the usage note: “an iterative process of biological discovery where our predictive model is used for hypothesis generation” is quite vague. Presumably there could be hundreds of thousands of hypotheses that could be tested. What classes of hypotheses do the authors propose? Is it possible to provide a specific example about what the authors envision?

We added the following example: ”where our predictive model is used for hypothesis generation (for example linking particular image features to particular gene regulatory modules)”. Linking single-cell imaging and sequencing as proposed in our work is particularly critical for applications where one of the data modalities is difficult/expensive to obtain. In many applications large-scale imaging datasets are relatively easier to obtain, including large-scale tissue biopsies or during early development of an organism. Our method can provide hypotheses based on the single-cell images on key regulatory genes that underlie disease progression or are regulated spatio-temporally during early development.

3. The authors discuss, but do not test, the addition of single-modality-specific latent features (Ni) to overcome the concern regarding single-modality-specific information. Is this indeed the case? If so, the authors should note that this feature of the architecture was not thoroughly vetted.

This is correct. We have added the following line in our discussion section to clarify: ”Empirically validating this aspect of our model is a potential direction for future work.”

4. The method details for the differential gene expression / Gene Ontology enrichment analyses are still lacking. Specifically, what GO software was used? What was the gene background used to determine GO enrichment? Was all of GO included or only a subset? What method was used and what was the cutoff in determining differential expression?

The main details for differential expression and gene ontology analysis are located in the

subsection "Gene expression analysis of naive CD4+ T-cells" under Methods in the main paper. We here include the relevant excerpts from that section regarding differential expression and GO enrichment:

"we performed differential expression analysis on the two subpopulations of naive CD4+ T-cells using Wilcoxon rank sum test. We defined marker genes as all genes with Bonferroni-corrected p-value of < 0.05 ... Gene ontology analysis was performed on these marker genes overexpressed in each cluster (average log-fold change > 0) using g:Profiler [37,38], keeping the top 5 gene ontology biological process terms with lowest p-values (Fig. 3d). All reported p-values (after adjusting for multiple hypothesis testing using the Benjamini–Hochberg procedure) were ≤ 0.05 ."

To provide more details in response to the reviewers questions: For GO enrichment we used g:Profiler [37,38] with the default options. This included using the "only annotated genes" option, which includes only genes with at least one annotation for the gene background. The whole GO graph (biological process) was queried for enrichment of terms. We reported the top 5 gene ontology biological process terms with lowest p-values. The only non-default change that we made was the use of the Benjamini–Hochberg procedure to adjust for multiple-hypothesis testing instead of using the custom method provided by g:Profiler. We used Wilcoxon rank sum test for differential expression analysis and used a cutoff of < 0.05 for the adjusted p-values.

We changed the following sentence in the revised manuscript to include more details as suggested by the reviewer: "Gene ontology analysis was performed on these marker genes overexpressed in each cluster (average log-fold change > 0) using g:Profiler [37,38] using the "only annotated genes" option, keeping the top 5 gene ontology biological process terms with lowest p-values (Fig. 3d).

This is a signed review. Gregory Way, Ph.D. Postdoctoral Associate Imaging Platform Broad Institute of MIT and Harvard 415 Main Street Cambridge, MA 02140

We thank the reviewer for the perceptive comments that helped improve our manuscript.

Reviewer #3 Comments and Responses

The authors have addressed all the issues that I raised in my first review. In particular, they discuss conditions and limitations of the presented method, which will be very valuable for the scientific community. I therefore believe that this paper is going to make a fine contribution to the field.

We thank the reviewer for the positive comments.

Figure 2: (a) Receiver Operating Characteristic (ROC) curve illustrating performance of classifiers trained to distinguish between peripheral and central chromatin patterns in images when evaluated on images translated from RNA-seq data. High performance of classifiers indicates that the alignment of the clusters in the latent space also holds in the original gene expression and imaging spaces and is robust to different architecture choices. The dotted line represents random guessing based on evenly-distributed classes. (b) ROC curves illustrating performance of classifiers trained to distinguish between quiescent and poised gene expression programs when evaluated on RNA-seq data translated from images. (c) Linear Discriminant Analysis (LDA) plots of single-cell RNA-seq (left) and imaging (right) datasets embedded in the latent space for models with different numbers of latent dimensions. The clusters with more quiescent (blue) and poised (green) gene expression programs from the RNA-seq dataset are aligned with the clusters with peripheral (blue) and central (green) chromatin patterns from the imaging dataset. (d) Same as (c), for models with different numbers of layers in the RNA-seq VAE. (e) Same as (c), for model with fully connected image VAE. Note that the model with latent dimension of 128 is the same model as the one with 4 layers in the RNA-seq VAE.

Figure 3: Differential gene expression analysis between cells with central and peripheral chromatin pattern performed on the predicted gene expression matrix translated from images using our methodology with different architecture choices. The predicted fold-change of gene expression based on images is strongly correlated with the observed fold-change of gene expression between quiescent and poised naive T-cells from the actual RNA-seq dataset. (a) Original model with 128 latent dimensions and 4 layers in the RNA-seq VAE, (b) model with 256 latent dimensions, (c) model with 3 layers in the RNA-seq VAE, (d) model with 5 layers in the RNA-seq VAE, (e) model with fully-connected Image VAE instead of convolutional.

REVIEWERS' COMMENTS

Reviewer #1 (Remarks to the Author):

The authors have satisfactorily addressed my comments. In particular I feel that the comparisons and placement of work in context is critical for a machine learning contribution. I would further ask that the authors discuss the fact that their model, like cycleGAN/MAGAN does not need paired data, and highlight situations where paired data is unavailable.

Reviewer #1: Comments and Responses

The authors have satisfactorily addressed my comments. In particular I feel that the comparisons and placement of work in context is critical for a machine learning contribution. I would further ask that the authors discuss the fact that their model, like cycleGAN/MAGAN does not need paired data, and highlight situations where paired data is unavailable.

We thank the reviewer for this suggestion. We now include the following sentences in the main text: "Similar to CycleGAN, our cross-modal autoencoder does not require paired samples, which is advantageous for many modalities, where the process of data collection results in destruction of the cell (e.g. RNA-seq) and thus the same cell cannot be used in another assay to measure a different modality (e.g. imaging). However, if additional information is available such as shared markers measured in all modalities and/or paired data, similar to the MAGAN approach, this prior information can be incorporated through additional terms in the loss function (see section on incorporating prior knowledge)."